# TOWARDS MORE THEORETICALLY-GROUNDED PARTICLE OPTIMIZATION SAMPLING FOR DEEP LEARNING

## ABSTRACT

Many deep-learning based methods such as Bayesian deep learning (DL) and deep reinforcement learning (RL) have heavily relied on the ability of a model being able to efficiently explore via Bayesian sampling. Particle-optimization sampling (POS) is a recently developed technique to generate high-quality samples from a target distribution by iteratively updating a set of interactive particles, with a representative algorithm the Stein variational gradient descent (SVGD). Though obtaining significant empirical success, the *non-asymptotic* convergence behavior of SVGD remains unknown. In this paper, we generalize POS to a stochasticity setting by injecting random noise in particle updates, called stochastic particle-optimization sampling (SPOS). Notably, for the first time, we develop *non-asymptotic convergence theory* for the SPOS framework, characterizing convergence of a sample approximation w.r.t. the number of particles and iterations under both convex- and noncovex-energy-function settings. Interestingly, we provide theoretical understanding of a pitfall of SVGD that can be avoided in the proposed SPOS framework, *i.e.*, particles tend to collapse to a local mode in SVGD under some particular conditions. Our theory is based on the analysis of nonlinear stochastic differential equations, which serves as an extension and a complementary development to the asymptotic convergence theory for SVGD such as (Liu, 2017). With such theoretical guarantees, SPOS can be safely and effectively applied on both Bayesian DL and deep RL tasks. Extensive results demonstrate the effectiveness of our proposed framework.

## 1 INTRODUCTION

Recent development of deep-learning techniques have required the ability of an algorithm to efficiently explore some particular space (*e.g.*, parameter space) via Bayesian sampling, due to high model complexity of modern deep models. Meanwhile, recent years have seen the development of scalable sampling methods such as stochastic gradient MCMC (SG-MCMC) (Welling & Teh, 2011; Chen et al., 2014; Ding et al., 2014; Chen et al., 2015) and Stein variational gradient descent (SVGD) (Liu & Wang, 2016) to facilitate big-data analysis. SG-MCMC is a family of scalable Bayesian sampling algorithms relying on Itó diffusions, linear stochastic differential equations (SDE) with appropriately designed coefficients such that the corresponding stationary distributions match a target distribution. One potential issue is that samples might be highly correlated partially due to the nature of Markov chains, leading to low sample efficiency, an undesirable property of SG-MCMC.

SVGD, on the other hand, belongs to the family of particle-based sampling methods, which optimize a set of interacting particles to minimize some distance metric between the target distribution and the distribution induced by the particles. By optimizing the distance measure, one maintains an optimal set of particles at each time. Recent development of SVGD has shown that the underlying mathematical principle is based on a family of *nonlinear* SDEs, in the sense that coefficients of the SDE depend on the current density of the particles. Though achieving numerous practical successes (Liu & Wang, 2016; Feng et al., 2017; Liu et al., 2017; Haarnoja et al., 2017; Liu & Zhu, 2018), little theory has been developed to understand the convergence property of the algorithm. A recent theoretical development has interpreted SVGD as a special type of gradient flow in the space of probability measures, and developed theory to disclose its *asymptotic* convergence behavior (Liu, 2017).

Recently, Chen et al. (2018) unified SG-MCMC and SVGD by proposing a particle-optimization sampling framework that interprets both as Wasserstein gradient flows (WGFs). Generally speaking, a WGF is a partial differential equation (PDE) defined on the space of probability measures, describing the evolution of a density function over time. In (Chen et al., 2018), the authors define a WGF by combining the corresponding Fokker-Planck equations for both SG-MCMC and SVGD, and solve it with deterministic particle approximations. However, due to the diffusion nature, deterministic-particle approximation leads to a hard-to-control error, challenging for theoretical analysis.

Based on the unified framework in (Chen et al., 2018), we propose to solve WGFs with *stochastic* particle approximation, leading to stochastic particle-optimization sampling (SPOS). The idea is instead of solving the WGF with an uncontrollable deterministic approximation for a diffusion term, we solve it stochastically by injecting random noise to the particle-update equations. Remarkably, for the first time, we develop nonasymptotic convergence theory for the family of SPOS algorithms, considering both convex- and nonconvex-energy functions. Different from existing theory for SG-MCMC based algorithms (Teh et al., 2016; Vollmer et al., 2016; Chen et al., 2015; Raginsky et al., 2017; Zhang et al., 2017; Xu et al., 2018), our development relies on the theory of *nonlinear SDEs*, which is more involved and less explored in literature. Particularly, we adopt tools from granular media equations (Malrieu, 2003; Cattiaux et al., 2008) to develop nonasymptotic error bounds in terms of the 1-Wasserstein distance. Please refer to Section M in the Supplementary Material (SM) for detailed distinctions of our work to existing work. Within our theoretical framework, we provide a formal theoretical understanding of a pitfall of SVGD, *i.e.,* particles tend to collapse to one point under some particular conditions; whereas this can be avoided in the proposed SPOS framework due to the injected random noise. Our theory is general enough to be applied for various deep-learning tasks, including Bayesian deep learning and Bayesian exploration in reinforcement learning. Our extensive experimental results well suggest advantages of our framework compared to existing methods.

## 2 PRELIMINARIES

This section introduces necessary preliminaries, along with notations used in this paper. For the sake of clarity, through out the paper, we use bold letters to denote variables in continuous-time diffusions and model definitions, *e.g.*, $\boldsymbol{\theta}_t$ in (1) defined below (indexed by "time" $t$). By contrast, normal unbold letters are used to denote parameters in algorithms (discrete solutions of continuous-time diffusions), *e.g.*, $\theta_k^{(i)}$ in (3) below (indexed by "iteration" $k$). For conciseness, all the proofs as well as some extra experimental results are presented in the SM. Discussion on the complexity of our algorithm is also included in Section L of the SM.

### 2.1 STOCHASTIC GRADIENT MCMC

In Bayesian sampling, we aim to generate random samples from a posterior distribution $p(\boldsymbol{\theta}|\mathcal{X}) \propto p(\mathcal{X}|\boldsymbol{\theta})p(\boldsymbol{\theta})$, where $\boldsymbol{\theta} \in \mathbb{R}^d$ represents the model parameter with a prior distribution $p(\boldsymbol{\theta})$, and $\mathcal{X} \triangleq \{\mathbf{x}_q\}_{q=1}^N$ represents the observed data with likelihood $p(\mathcal{X}|\boldsymbol{\theta}) = \prod_q p(\mathbf{x}_q|\boldsymbol{\theta})$. Define the potential energy as: $U(\boldsymbol{\theta}) \triangleq -\log p(\mathcal{X}|\boldsymbol{\theta}) - \log p(\boldsymbol{\theta}) = -\sum_{q=1}^N \left(\log p(\mathbf{x}_q|\boldsymbol{\theta}) + \frac{1}{N}\log p(\boldsymbol{\theta})\right) \triangleq \sum_{q=1}^N U_q(\boldsymbol{\theta})$. SG-MCMC algorithms belong to diffusion-based sampling methods, where a continuous-time diffusion process is designed such that its stationary distribution matches the target posterior distribution. The diffusion process is driven by a specific stochastic differential equation (SDE). For example, in stochastic gradient Langevin dynamic (SGLD), the SDE endows the following form:

$$\mathrm{d}\boldsymbol{\theta}_t = -\beta^{-1}F(\boldsymbol{\theta}_t)\mathrm{d}t + \sqrt{2/\beta}\mathrm{d}\mathcal{W}_t \,, \tag{1}$$

where $F(\boldsymbol{\theta}) \triangleq \nabla_{\boldsymbol{\theta}}U(\boldsymbol{\theta}) = \sum_{q=1}^N \nabla_{\boldsymbol{\theta}}U_q(\boldsymbol{\theta}) \triangleq \sum_{q=1}^N F_q(\boldsymbol{\theta})$; $t$ is the time index; $\beta > 0$ is the temperature parameter; and $\mathcal{W}_t \in \mathbb{R}^d$ is a $d$-dimensional Brownian motion. More instances of SDEs corresponding to other SG-MCMC algorithms can be defined by specifying different forms of $F$ and potentially other diffusion coefficients. We focus on SGLD and (1) in this paper, and refer interested readers to (Ma et al., 2015) for more detailed description of general SG-MCMC algorithms. Denote the probability density function of $\boldsymbol{\theta}_t$ in (1) as $\nu_t$, and $\mathbf{a} \cdot \mathbf{b} \triangleq \mathbf{a}^\top \mathbf{b}$ for two vectors $\mathbf{a}$ and $\mathbf{b}$. It is known that $\nu_t$ is characterized by the following Fokker-Planck (FP) equation (Risken, 1989):

$$\partial_t \nu_t = \nabla_{\boldsymbol{\theta}} \cdot (\beta^{-1}\nu_t F(\boldsymbol{\theta}) + \beta^{-1}\nabla_{\boldsymbol{\theta}}\nu_t) \,. \tag{2}$$

where the stationary distribution $\nu_\infty$ equals to our target distribution $p(\boldsymbol{\theta}|\mathcal{X})$ according to Chiang & Hwang (1987). SGLD generates samples from $p(\boldsymbol{\theta}|\mathcal{X})$ by numerically solving the SDE (1). For

scalability, it replaces $F(\theta_k)$ in each iteration with an unbiased evaluation by randomly sampling a subset of $\mathcal{X}$, $i.e.$ $F(\theta_k)$ is approximated by: $G_k \triangleq \frac{N}{B_k} \sum_{q \in \mathcal{I}_k} F_q(\theta_k)$, where $\mathcal{I}_k$ is a random subset of $[1, 2, \cdots, N]$ with size $B_k$ in each iteration. Based on the above settings, SGLD uses the Euler method with stepsize $h_k$ to numerically solve (1) and obtains the update equation: $\theta_{k+1} = \theta_k - \beta^{-1} G_k h_k + \sqrt{2\beta^{-1} h_k} \xi_k$, $\xi_k \sim \mathcal{N}(\mathbf{0}, \mathbf{I})$.

## 2.2 STEIN VARIATIONAL GRADIENT DESCENT

Different from SG-MCMC, SVGD is a deterministic particle-optimization algorithm that is able to generate samples from a target distribution. In the algorithm, a set of particles interact with each other, driving them to high density locations in the parameter space while keeping them far away from each other with *repulsive* force. The update equations of the particles follow the fastest descent direction of the KL-divergence between current empirical distribution of the particles and the target distribution, on an RKHS induced by a kernel function $\kappa(\cdot, \cdot)$ (Liu & Wang, 2016). Formally, Liu & Wang (2016) derived the following updating rules for the particles $\{\theta_k^{(i)}\}_{i=1}^M$ at the $k$-th iteration with stepsize $h_k$ and $G_k^{(i)} \triangleq \frac{N}{B_k} \sum_{q \in \mathcal{I}_k} F_q(\theta_k^{(i)})$:

$$\theta_{k+1}^{(i)} = \theta_k^{(i)} + \frac{h_k}{M} \sum_{j=1}^M \left[ \kappa(\theta_k^{(j)}, \theta_k^{(i)}) G_k^{(i)} + \nabla_{\theta_k^{(j)}} \kappa(\theta_k^{(j)}, \theta_k^{(i)}) \right], \quad \forall i \tag{3}$$

where the first term in the bracket encourages particles to locate on high density modes, and the second term serves as repulsive force that pushes away different particles. Different from SG-MCMC, only particles at the *current* iteration, $\{\theta_k^{(i)}\}_{i=1}^M$, are used to approximate the target distribution.

## 2.3 PARTICLE-OPTIMIZATION BASED SAMPLING METHODS

SG-MCMC and SVGD, though look closely related, behave very differently in terms of algorithms, *e.g.*, stochastic and noninteractive versus deterministic and interactive particle updates. Recently, Chen et al. (2018) proposed a deterministic particle-optimization framework that unifies both SG-MCMC and SVGD. Specifically, the authors viewed both SG-MCMC and SVGD as Wasserstein gradient flows (WGFs) on the space of probabilistic measures, and derived several deterministic particle-optimization techniques for particle evolutions, like what SVGD does. For SG-MCMC, the FP equation (2) for SGLD is a special type of WGFs. Together with an interpretation of SVGD as a special case of the Vlasov equation in nonlinear PDE literature, Chen et al. (2018) proposed a general form of PDE to characterize the evolution of the density for the model parameter $\boldsymbol{\theta}$, denoted as $\nu_t$ at time $t$ with $\nu_\infty$ matching our target (posterior) distribution, *i.e.*,

$$\partial_t \nu_t = \nabla_{\boldsymbol{\theta}} \cdot \left( \nu_t \beta^{-1} F(\boldsymbol{\theta}) + \nu_t (\mathcal{K} * \nu_t(\boldsymbol{\theta})) + \beta^{-1} \nabla_{\boldsymbol{\theta}} \nu_t \right), \tag{4}$$

where $\mathcal{K}$ is a function controlling the interaction of particles in the PDE system. For example, in SVGD, Chen et al. (2018) showed that $\mathcal{K}$ and $\mathcal{K} * \nu_t(\boldsymbol{\theta})$ endow the following forms:

$$\mathcal{K}(\boldsymbol{\theta}, \boldsymbol{\theta}') \triangleq F(\boldsymbol{\theta}') \kappa(\boldsymbol{\theta}', \boldsymbol{\theta}) - \nabla_{\boldsymbol{\theta}'} \kappa(\boldsymbol{\theta}', \boldsymbol{\theta}) \; and \; \mathcal{K} * \nu_t(\boldsymbol{\theta}) \triangleq \int \mathcal{K}(\boldsymbol{\theta}, \boldsymbol{\theta}') \nu_t(\boldsymbol{\theta}') d\boldsymbol{\theta}' \tag{5}$$

where $\kappa(\cdot, \cdot)$ is a kernel function such as the RBF kernel. In the following, we introduce a new unary function $K(\boldsymbol{\theta}) = \exp(-\frac{\|\boldsymbol{\theta}\|^2}{\eta^2})$, thus $\kappa(\boldsymbol{\theta}, \boldsymbol{\theta}')$ can be rewritten as $\kappa(\boldsymbol{\theta}, \boldsymbol{\theta}') = K(\boldsymbol{\theta} - \boldsymbol{\theta}')$. Hence, (4) with $\mathcal{K}$ defined in (5) can be equivalently rewritten as:

$$\partial_t \nu_t = \nabla_{\boldsymbol{\theta}} \cdot \left( \nu_t \beta^{-1} F(\boldsymbol{\theta}) + \nu_t (E_{Y \sim \nu_t} K(\boldsymbol{\theta} - Y) F(Y) - \nabla K * \nu_t(\boldsymbol{\theta})) + \beta^{-1} \nabla_{\boldsymbol{\theta}} \nu_t \right), \tag{6}$$

where $Y$ is a random sample from $\nu_t$ but independent of $\boldsymbol{\theta}$. Importantly,

**Proposition 1 (Chen et al. (2018))** *The stationary distribution of* (6) *equals to our target distribution, which means* $\nu_\infty(\boldsymbol{\theta}) = p(\boldsymbol{\theta} | \mathcal{X})$.

Chen et al. (2018) proposed to solve (4) numerically with deterministic particle-optimization algorithms such as the blob method. Specifically, the continuous density $\nu_t$ is approximated by a set of $M$ particles $\{\boldsymbol{\theta}_t^{(i)}\}_{i=1}^M$ that evolve over time $t$, *i.e.* $\nu_t \approx \frac{1}{M} \sum_{i=1}^M \delta_{\boldsymbol{\theta}_t^{(i)}}(\boldsymbol{\theta})$, where $\delta_{\boldsymbol{\theta}_t^{(i)}}(\boldsymbol{\theta}) = 1$ if $\boldsymbol{\theta} = \boldsymbol{\theta}_t^{(i)}$ and 0 otherwise. Note $\nabla_{\boldsymbol{\theta}} \nu_t$ in (4) is no longer a valid definition when adopting particle approximation for $\nu_t$. Consequently, $\nabla_{\boldsymbol{\theta}} \nu_t$ needs nontrivial approximations, *e.g.*, by discrete gradient flows or blob methods proposed in (Chen et al., 2018). We omit the details here for simplicity.

## 3 STOCHASTIC PARTICLE-OPTIMIZATION SAMPLING ALGORITHMS

The deterministic particle-approximation methods proposed by Chen et al. (2018) to approximately solve the WGF problem (4) introduce approximation errors for $\nabla_{\boldsymbol{\theta}} \nu_t$ that are hard to control analytically. To overcome this problem, we propose to solve (4) *stochastically* to replace the $\nabla_{\boldsymbol{\theta}} \nu_t$ term with a Brownian motion. Specifically, first note that the term $\beta^{-1} \nabla_{\boldsymbol{\theta}} \cdot \nabla_{\boldsymbol{\theta}} \nu_t$ is contributed from Brownian motion, *i.e.*, solving the SDE, $\mathrm{d}\boldsymbol{\theta}_t = \sqrt{2\beta^{-1}} \mathrm{d}\mathcal{W}_t$, is equivalent to solving the corresponding FP equation: $\partial \nu_t = \beta^{-1} \nabla_{\boldsymbol{\theta}} \cdot \nabla_{\boldsymbol{\theta}} \nu_t$. Consequently, we decompose RHS of (4) into two parts: $F_1 \triangleq \nabla_{\boldsymbol{\theta}} \cdot (\nu_t \beta^{-1} F(\boldsymbol{\theta}_t) + (\mathcal{K} * \nu_t)\nu_t)$ and $F_2 \triangleq \beta^{-1} \nabla_{\boldsymbol{\theta}} \cdot \nabla_{\boldsymbol{\theta}} \nu_t$. Our idea is to solve $F_1$ deterministically under a PDE setting, and solve $F_2$ stochastically based on its corresponding SDE. When adopting particle approximation for the density $\nu_t$, both solutions of $F_1$ and $F_2$ are represented in terms of particles $\{\boldsymbol{\theta}_t^{(i)}\}$. Thus we can combine the solutions from the two parts directly to approximate the original exact solution of (4). Similar to the results of SVGD in Section 3.3 in (Liu, 2017), we first formally show in Theorem 2 that when approximating $\nu_t$ with particles, *i.e.*, $\nu_t \approx \frac{1}{M} \sum_{i=1}^{M} \delta_{\boldsymbol{\theta}_t^{(i)}}(\boldsymbol{\theta})$, the PDE can be transformed into a system of deterministic differential equations with interacting particles.

**Theorem 2** *When approximating $\nu_t$ in (4) with particles $\{\boldsymbol{\theta}_t^{(i)}\}_i$, the PDE $\partial_t \nu_t = F_1$ reduces to the following system of differential equations describing evolutions of the particles over time:* $\forall i$

$$\mathrm{d}\boldsymbol{\theta}_t^{(i)} = -\beta^{-1} F(\boldsymbol{\theta}_t^{(i)})\mathrm{d}t - \frac{1}{M} \sum_{j=1}^{M} K(\boldsymbol{\theta}_t^{(i)} - \boldsymbol{\theta}_t^{(j)})F(\boldsymbol{\theta}_t^{(j)})\mathrm{d}t + \frac{1}{M} \sum_{j=1}^{M} \nabla K(\boldsymbol{\theta}_t^{(i)} - \boldsymbol{\theta}_t^{(j)})\mathrm{d}t \quad (7)$$

On the other hand, by solving $\partial_t \nu_t = F_2$ stochastically with its equivalent SDE counterpart, we arrive at the following differential equation system, describing evolutions of the particles $\{\boldsymbol{\theta}_t^{(i)}\}$ over time $t$: $\forall i$

$$\mathrm{d}\boldsymbol{\theta}_t^{(i)} = \left( -\beta^{-1} F(\boldsymbol{\theta}_t^{(i)}) - \frac{1}{M} \sum_{j=1}^{M} K(\boldsymbol{\theta}_t^{(i)} - \boldsymbol{\theta}_t^{(j)})F(\boldsymbol{\theta}_t^{(j)}) + \frac{1}{M} \sum_{j=1}^{M} \nabla K(\boldsymbol{\theta}_t^{(i)} - \boldsymbol{\theta}_t^{(j)}) \right) \mathrm{d}t + \sqrt{2\beta^{-1}} \mathrm{d}\mathcal{W}_t^{(i)}$$
$$(8)$$

Our intuition is that if the particle evolution (8) can be solved exactly, the solution of (6) $\nu_t$ will be well-approximated by the particles $\{\boldsymbol{\theta}_t^{(i)}\}_{i=1}^{M}$. In our theory, we show this intuition is actu-

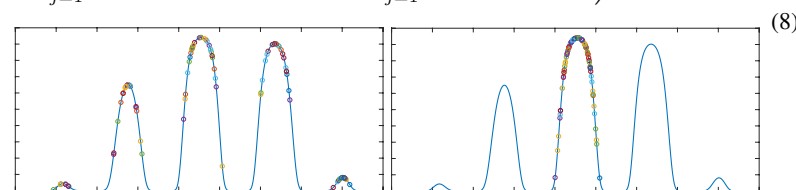

Figure 1: Comparison of SPOS (left) and SVGD (right) on a multi-mode distribution. The circles with different colors are the resulting 100 particles, which are able to spread over all modes for SPOS.

ally true. In practice, however, solving (8) is typically infeasible, and thus numerical methods are adopted. Furthermore, in the case of big data, following SG-MCMC, $F(\theta_k^{(i)})$ is typically replaced by a stochastic version $G_k^{(i)} \triangleq \frac{N}{B_k} \sum_{q \in \mathcal{I}_k} F_q(\theta_k^{(i)})$ evaluated with a minibatch of data of size $B_k$ for computational feasibility. Based on the Euler method (Chen et al., 2015) with a stepsize $h_k$, (8) leads to the following updates for the particles at the $k$-th iteration

$$\theta_{k+1}^{(i)} = \theta_k^{(i)} - h_k \beta^{-1} G_k^{(i)} - \frac{h_k}{M} \sum_{j=1}^{M} K(\theta_k^{(i)} - \theta_k^{(j)})G_k^{(j)}$$

$$+ \frac{h_k}{M} \sum_{j=1}^{M} \nabla K(\theta_k^{(i)} - \theta_k^{(j)}) + \sqrt{2\beta^{-1}h_k}\xi_k^{(i)}, \xi_k^{(i)} \sim \mathcal{N}(\mathbf{0}, \mathbf{I}) \quad \forall i . \quad (9)$$

We called the algorithm with particle update equations (9) stochastic particle-optimization sampling (described in Algorithm 3), in the sense that particles are optimized stochastically with extra random Gaussian noise. Intuitively, the added Gaussian noise would enhance the ability of the algorithm to jump out of local modes, leading to better ergodic properties compared to standard SVGD. This serves as one of our motivations to generalize SVGD to SPOS. To illustrate the advantage of introducing the noise term, we compare SPOS and SVGD on sampling a difficult multi-mode distribution, with the density function given in Section A of the SM. The particles are initialized on a local mode close to zero. Figure 1 plots the final locations of the particles along with the true density, which shows that particles in SPOS are able to reach different modes, while they are all trapped at one mode in SVGD. This *pitfall* of SVGD will be studied formally in Section 4.4.

## 4    NON-ASYMPTOTIC CONVERGENCE ANALYSIS: THE CONVEX CASE

In this section, we develop non-asymptotic convergence theory for the proposed SPOS when the energy function $U(\boldsymbol{\theta})$ is convex. The nonconvex case is discussed in Section 5. We prove non-asymptotic convergence rates for SPOS algorithm under the 1-Wasserstein metric $W_1$, a special case of p-Wasserstein metric de-

---

**Algorithm 1** Stochastic Particle-Optimization Sampling

**Input:** Initial particles $\{\theta_0^{(i)}\}_{i=1}^M$ with $\theta_0^{(i)} \in \mathbb{R}^d$, step size $h_k$, batch size $B_k$
  1: **for** iteration $k$= 0,1,...,T **do**
  2:    Update $\theta_{k+1}^{(i)}$ with (9) for $\forall i$.
  3: **end for**
**Output:** $\{\theta_T^{(i)}\}_{i=1}^M$

---

fined as $W_p(\mu,\nu) = \left( \inf_{\zeta \in \Gamma(\mu,\nu)} \int_{\mathbb{R}^d \times \mathbb{R}^d} \|X_\mu - X_\nu\|^p d\zeta(X_\mu, X_\nu) \right)^{1/p}$, where $\Gamma(\mu,\nu)$ is the set of joint distributions on $\mathbb{R}^d \times \mathbb{R}^d$ with marginal distribution $\mu$ and $\nu$. Note that SPOS reduces to SVGD when $\beta \to \infty$, thus our theory also sheds light on the convergence behavior of SVGD, where non-asymptotic theory is currently missing, despite the asymptotic theory developed recently (Liu, 2017; Lu et al., 2018). It is worth noting that part of our proofs are generalization of techniques for analyzing granular media equations in (Malrieu, 2003; Cattiaux et al., 2008).

### 4.1    BASIC SETUP AND ASSUMPTIONS

Due to the exchangeability of the particle system $\{\boldsymbol{\theta}_t^{(i)}\}_{i=1}^M$ in (8), if we initialize all the particles $\boldsymbol{\theta}_t^{(i)}$ with the same distribution $\rho_0$, they would endow the same distribution for each time $t$. We denote the distribution of each $\boldsymbol{\theta}_t^{(i)}$ as $\rho_t$. Similar arguments hold for the particle system $\{\theta_k^{(i)}\}_{i=1}^M$ in (9), and thus we denote the distribution of each $\theta_k^{(i)}$ as $\mu_k$. To this end, our analysis aims at bounding $W_1(\mu_T, \nu_\infty)$ since $\nu_\infty$ equals to our target distribution $p(\boldsymbol{\theta}|\mathcal{X})$ according to Proposition 1. Before proceeding to our theoretical results, we first present the following basic assumptions.

**Assumption 1** *Assume $F$ and $K$ satisfy the following conditions:*

- *There exist positive $m_F$ and $m_K$, such that $\langle F(\boldsymbol{\theta}) - F(\boldsymbol{\theta}'), \boldsymbol{\theta} - \boldsymbol{\theta}' \rangle \geq m_F \|\boldsymbol{\theta} - \boldsymbol{\theta}'\|^2$ and $\langle \nabla K(\boldsymbol{\theta}) - \nabla K(\boldsymbol{\theta}'), \boldsymbol{\theta} - \boldsymbol{\theta}' \rangle \leq -m_K \|\boldsymbol{\theta} - \boldsymbol{\theta}'\|^2$.*

- *$F$ is bounded by $H_F$ and $L_F$-Lipschitz continuous i.e., $\|F(\boldsymbol{\theta})\| \leq H_F$ and $\|F(\boldsymbol{\theta}_1) - F(\boldsymbol{\theta}_2)\| \leq L_F \|\boldsymbol{\theta}_1 - \boldsymbol{\theta}_2\|$; .*

- *$K$ is $L_K$-Lipschitz continuous; $\nabla K$ is bounded by $H_{\nabla K}$ and $L_{\nabla K}$-Lipschitz continuous*

- *$F(\mathbf{0}) = \mathbf{0}$ and $K$ is an even function, i.e., $K(-\boldsymbol{\theta}) = K(\boldsymbol{\theta})$.*

Note the first bullet indicates $U$ to be a convex function and $K$ to be a concave function. For an RBF kernel, the later could be achieved by setting the bandwidth large enough and only considering the concave region for simplicity. This assumption is used for revealing some undesired property of SVGD developed below. We do not need such an assumption when analyzing under a nonconvex energy function $U$ in Section 5. Then "$F(\mathbf{0}) = \mathbf{0}$" in the second bullet is reasonable, as $F$ in our setting corresponds to an unnormalized log-posterior, which can be shifted such that $F(\mathbf{0}) = \mathbf{0}$ for a specific problem. Since we often care about bounded space in practice, we can realize the third bullet due to the continuity of $K$ and $\nabla K$.

The high-level idea of bounding $W_1(\mu_T, \nu_\infty)$ in this section is to decompose it as follows:

$$W_1(\mu_T, \nu_\infty) \leq W_1(\mu_T, \rho_{\sum_{k=0}^{T-1} h_k}) + W_1(\rho_{\sum_{k=0}^{T-1} h_k}, \nu_{\sum_{k=0}^{T-1} h_k}) + W_1(\nu_{\sum_{k=0}^{T-1} h_k}, \nu_\infty). \quad (10)$$

### 4.2    BOUNDS WITH STOCHASTIC PARTICLE APPROXIMATION

We firstly bound $W_1(\rho_{\sum_{k=0}^{T-1} h_k}, \nu_{\sum_{k=0}^{T-1} h_k})$ and $W_1(\nu_{\sum_{k=0}^{T-1} h_k}, \nu_\infty)$ with the following theorems.

**Theorem 3** *Under Assumption 1 and let $\rho_0 = \nu_0$, there exist some positive constants $c_1$ and $c_2$ independent of $(M, t)$ and satisfying $c_2 < \beta^{-1}$ such that*

$$W_1(\rho_t, \nu_t) \leq c_1(\beta^{-1} - c_2)^{-1} M^{-1/2}, \quad \forall t. \quad (11)$$

**Remark 1** *According to Theorem 3, we can bound the $W_1(\rho_{\sum_{k=0}^{T-1} h_k}, \nu_{\sum_{k=0}^{T-1} h_k})$ term as $W_1(\rho_{\sum_{k=0}^{T-1} h_k}, \nu_{\sum_{k=0}^{T-1} h_k}) \leq \frac{c_1}{\sqrt{M}(\beta^{-1}-c_2)}$. Furthermore, by letting $t \to \infty$, we have $W_1(\rho_\infty, \nu_\infty) \leq \frac{c_1}{\sqrt{M}(\beta^{-1}-c_2)}$, which is an important intermediate result to prove the following theorem.*

**Theorem 4** *Under Assumption 1, the following holds: $W_1(\nu_t, \nu_\infty) \leq c_3 e^{-2\lambda_1 t}$, where $\lambda_1 = \beta^{-1} m_F - 3H_F L_K - 2L_F$ and $c_3$ is some positive constant independent of $(M, t)$. Hence, the $W_1(\nu_{\sum_{k=0}^{T} h_k}, \nu_\infty)$ term in (10) can be bounded as:*

$$W_1(\nu_{\sum_{k=0}^{T-1} h_k}, \nu_\infty) \leq c_3 \exp\left(-2\lambda_1 \left(\sum_{k=0}^{T-1} h_k\right)\right) . \tag{12}$$

To ensure $W_1(\nu_{\sum_{k=0}^{T-1} h_k}, \nu_\infty)$ to decrease over time, one needs to choose $\beta$ small enough such that $\lambda_1 > 0$. This also sheds light on a failure case of SVGD (where $\beta \to \infty$) discussed in Section 4.4.

### 4.3 BOUNDS WITH A NUMERICAL SOLUTION

To bound the $W_1(\mu_T, \rho_{\sum_{k=0}^{T-1} h_k})$ term, we adopt techniques from (Raginsky et al., 2017; Xu et al., 2018) on analyzing the behaviors of SGLD, and derive the following results for our SPOS algorithm:

**Theorem 5** *Under Assumptions 1, for a fixed step size $h_k = h$ that is small enough, the corresponding $W_1(\mu_T, \rho_{Th})$ is bounded as*

$$W_1(\mu_T, \rho_{Th}) \leq c_4 M d^{\frac{3}{2}} \beta^{-3} (c_5 \beta^2 B^{-1} + c_6 h)^{\frac{1}{2}} T^{\frac{1}{2}} h^{\frac{1}{2}} \tag{13}$$

*where $B$ is the fixed size of the minibatch in each iteration and $(c_4, c_5, c_6)$ are some positive constants independent of $(M, T, h)$.*

The dependence of $T$ in the bound above makes the bound relatively loose. Fortunately, we can improve the bound to make it independent of $T$ by considering a decreasing-stepsize SPOS algorithm, stated in Theorem 6.

**Theorem 6** *Under Assumptions 1, for a decreasing step size $h_k = h_0/(k+1)$, and let the minibatch size in each iteration $k$ be $B_k = B_0 + [\log(k+1)]^{100/99}$, the corresponding $W_1(\mu_T, \rho_{\sum_{k=0}^{T-1} h_k})$ term is bounded, for some $\beta$ small enough, as*

$$W_1(\mu_T, \rho_{\sum_{k=0}^{T-1} h_k}) \leq c_4 \beta^{-3} M d^{\frac{3}{2}} (c_7 h_0^3 + c_8 \beta^3 h_0/B_0 + c_9 h_0^2 \beta^2)^{1/2} , \tag{14}$$

*where $B_0$ is the initial minibatch size, and $(c_4, c_7, c_8, c_9)$ are some positive constants independent of $(M, T, h_0)$.*

Note $B_k$ increases at a very low speed, *e.g.*, only by 15 after $10^5$ iterations, thus it would not affect algorithm efficiency. According to Theorem 6, $W_1(\mu_T, \rho_{\sum_{k=0}^{T-1} h_k})$ would approach zero when $h_0^{1/2} M \to 0$. By directly combining results from Theorem 3–6, one can easily bound the target $W_1(\mu_T, \nu_\infty)$. Detailed statements are given in Theorem 15–16 in Section H of the SM.

### 4.4 A PITFALL OF SVGD

Based on the above analysis, we now formally show a pitfall of SVGD, *i.e.*, particles in SVGD tend to collapse to a local mode under some particular conditions. Inspired by the work on analyzing the granular media equations by Malrieu (2003); Cattiaux et al. (2008), we measure this by calculating the expected distance between particles, called expected particle distance (EPD). Firstly, we bound the EPD for the proposed SPOS algorithm in Theorem 7.

**Theorem 7** *Under Assumption 1, further assuming every $\{\boldsymbol{\theta}_t^{(i)}\}$ with the same initial probability law $\rho_0$ and $\Gamma \triangleq \mathbb{E}_{\boldsymbol{\theta} \sim \rho_0, \boldsymbol{\theta}' \sim \rho_0}[\|\boldsymbol{\theta} - \boldsymbol{\theta}'\|^2] < \infty$. Choose a $\beta$ such that $\lambda = \frac{m_F}{\beta} + m_K - H_F L_K > 0$. Then the EPD of SPOS is bounded as: $EPD \triangleq \sqrt{\sum_{i,j}^{M} \mathbb{E}\|\boldsymbol{\theta}_t^{(i)} - \boldsymbol{\theta}_t^{(j)}\|^2} \leq C_1 e^{-2\lambda t} + 4\sqrt{\frac{d}{\beta}} \frac{M}{\lambda}$, where $C_1 = M(M-1)\Gamma - 4\sqrt{d\beta^{-1}} \frac{M}{\lambda}$.*

**Remark 2** *There are two interesting cases: i) When $C_1 > 0$, the EPD would decrease to the bound $4\sqrt{d\beta^{-1}} M/\lambda$ along time $t$. This represents the phenomenon of an attraction force between particles;*

*ii) When $C_1 < 0$, the EPD would increase to the same bound, which represents the phenomenon of a repulsive force between particles, e.g., when particles are initialized with the same value ($\Gamma = 0$), they would be pushed away from each other until the EPD increases to the aforementioned bound.*

Intuitively, the EPD for SVGD can be obtained by taking the $\beta \to \infty$ limit. Corollary 8 formally characterizes the particle-degeneracy phenomenon of SVGD, which has been empirically studied in (Zhuo et al., 2018).

**Corollary 8** *Under the same conditions of Theorem 7, the EPD in SVGD is bounded as: EPD $\triangleq$ $\sqrt{\sum_{i,j}^{M} \|\boldsymbol{\theta}_t^{(i)} - \boldsymbol{\theta}_t^{(j)}\|^2} \leq C_0 e^{-2\lambda t}$, where $C_0 = \sqrt{\sum_{i,j}^{M} \|\boldsymbol{\theta}_0^{(i)} - \boldsymbol{\theta}_0^{(j)}\|^2}$ and $\lambda = m_K - H_F L_K$.*

**Remark 3** *We would like to emphasize two points: 1) In the case of $\lambda \geq 0$, Corollary 8 indicates that particles in SVGD would collapse to a point when $t \to \infty$. In practice, we usually find that particles are trapped in a local mode instead of collapsing. This is due to two reasons: i) numerical errors inject noise into the particles; ii) some particles are out of the concave region of $K$ stated in Assumption 1 in SVGD, which is required for the theory to hold. All these make the empirical EPD not exactly the same as the true particle distance. 2) Corollary 8 also applies when the energy function is nonconvex. Our proof in the SM considers the nonconvex case as well. Consequently, this serves as a strong theoretical motivation to apply SPOS instead of SVGD in deep learning.*

## 5 NON-ASYMPTOTIC CONVERGENCE ANALYSIS: THE NONCONVEX CASE

Since the non-convex case is much more complicated than the convex case, we reply on different assumptions and adopt another distance metric, denoted as $\tilde{\mathcal{B}}$, to characterize the convergence behavior of SPOS under the non-convex case. Specifically, $\tilde{\mathcal{B}}(\mu, \nu)$ is defined as $\tilde{\mathcal{B}}(\mu, \nu) \triangleq |\mathbb{E}_{\boldsymbol{\theta} \sim \mu}[f(\boldsymbol{\theta})] - \mathbb{E}_{\boldsymbol{\theta} \sim \nu}[f(\boldsymbol{\theta})]|$ for a known $L_f$-continuous function $f$ satisfying Assumption 2 below. Note such metric has also been adopted in (Vollmer et al., 2016; Chen et al., 2015). Our analysis considers $(T, M, h_k)$ as variables in $\tilde{\mathcal{B}}$. In addition, we use $\{\hat{\theta}_k^{(i)}\}_{i=1}^{M}$ to denote the particles when full gradients are adopted in (9). The distribution of the particles is denoted as $\hat{\mu}_k$.

Our high-level idea of bounding $\tilde{\mathcal{B}}(\mu_T, \nu_\infty)$ is to decompose it as follows:

$$\tilde{\mathcal{B}}(\mu_T, \nu_\infty) \leq \tilde{\mathcal{B}}(\mu_T, \hat{\mu}_T) + \tilde{\mathcal{B}}(\hat{\mu}_T, \hat{\mu}_\infty) + \tilde{\mathcal{B}}(\hat{\mu}_\infty, \rho_\infty) + \tilde{\mathcal{B}}(\rho_\infty, \nu_\infty) \tag{15}$$

Our second idea is to concatenate the particles at each time into a single vector representation, *i.e.* defining the new parameter at time $t$ as $\boldsymbol{\Theta}_t \triangleq [\boldsymbol{\theta}_t^{(1)}, \cdots, \boldsymbol{\theta}_t^{(M)}] \in \mathbb{R}^{Md}$. Consequently, the nonlinear SDE system (8) can be turned into a linear SDE, which means $\boldsymbol{\Theta}_t$ is driven by the following linear SDE:

$$\mathrm{d}\boldsymbol{\Theta}_t = -F_{\boldsymbol{\Theta}}(\boldsymbol{\Theta}_t)\mathrm{d}t + \sqrt{2\beta^{-1}}\mathrm{d}\mathcal{W}_t^{(Md)} , \tag{16}$$

where $F_{\boldsymbol{\Theta}}(\boldsymbol{\Theta}_t) \triangleq [\beta^{-1}F(\boldsymbol{\theta}_t^{(1)}) - \frac{1}{M}\sum_{j=1}^{M}\nabla K(\boldsymbol{\theta}_t^{(1)} - \boldsymbol{\theta}_t^{(j)}) + \frac{1}{M}\sum_{j=1}^{M}K(\boldsymbol{\theta}_t^{(1)} - \boldsymbol{\theta}_t^{(j)})F(\boldsymbol{\theta}_t^{(j)}), \cdots, \beta^{-1}F(\boldsymbol{\theta}_t^{(M)}) - \frac{1}{M}\sum_{j=1}^{M}\nabla K(\boldsymbol{\theta}_t^{(M)} - \boldsymbol{\theta}_t^{(j)}) + \frac{1}{M}\sum_{j=1}^{M}K(\boldsymbol{\theta}_t^{(M)} - \boldsymbol{\theta}_t^{(j)})F(\boldsymbol{\theta}_t^{(j)})]$ is a vector function $\mathbb{R}^{Md} \to \mathbb{R}^{Md}$, and $\mathcal{W}_t^{(Md)}$ is Brownian motion of dimension $Md$. Similarly, we can define $\hat{\Theta}_k \triangleq [\hat{\theta}_k^{(1)}, \cdots, \hat{\theta}_k^{(M)}] \in \mathbb{R}^{Md}$ for the full-gradient case. Hence, it can be seen that through such a decomposition in (15), the bound related to a nonlinear SDE system (8) reduces to that of a linear SDE. The second term $\tilde{\mathcal{B}}(\hat{\mu}_T, \hat{\mu}_\infty)$ reflexes the geometric ergodicity of a linear dynamic system with a numerical method. It is known that even if a dynamic system has an exponential convergence rate to its equilibrium, its corresponding numerical method might not. Our bound for $\tilde{\mathcal{B}}(\hat{\mu}_T, \hat{\mu}_\infty)$ is essentially a specification of the result of Mattingly et al. (2002), which has also been applied by Xu et al. (2018). The third term $\tilde{\mathcal{B}}(\hat{\mu}_\infty, \rho_\infty)$ reflects the numerical error of a linear SDE, which has been studied in related literature such as (Chen et al., 2015). To this end, we adopt standard assumptions used in the analysis of SDEs (Vollmer et al., 2016; Chen et al., 2015), rephrased in Assumption 2.

**Assumption 2 (Assumption in (Vollmer et al., 2016; Chen et al., 2015))** *For the linear SDE (16) and a Lipschitz function $f$, let $\psi$ be the solution functional of the Poisson equation: $\mathcal{G}\psi(\hat{\Theta}_k) =$*

$\frac{1}{M}\sum_{i=1}^{M} f(\hat{\theta}_k^{(i)}) - \mathbb{E}_{\boldsymbol{\theta}\sim p(\boldsymbol{\theta}|\mathcal{D})}[f(\boldsymbol{\theta})]$, *where $\mathcal{G}$ denotes the infinite generator of the linear SDE (16). Assume $\psi$ and its up to 4th-order derivatives, $\mathcal{D}^k\psi$, are bounded by a function $\mathcal{V}$, i.e., $\|\mathcal{D}^k\psi\| \leq H_k\mathcal{V}^{p_k}$ for $k = (0,1,2,3,4)$, $H_k, p_k > 0$. Furthermore, the expectation of $\mathcal{V}$ on $\{\boldsymbol{\Theta}_t\}$ is bounded: $\sup_l \mathbb{E}\mathcal{V}^p(\boldsymbol{\Theta}_t) < \infty$, and $\mathcal{V}$ is smooth such that $\sup_{s\in(0,1)} \mathcal{V}^p(s\boldsymbol{\Theta} + (1-s)\boldsymbol{\Theta}') \leq H(\mathcal{V}^p(\boldsymbol{\Theta}) + \mathcal{V}^p(\boldsymbol{\Theta}'))$, $\forall \boldsymbol{\Theta}, \boldsymbol{\Theta}', p \leq \max\{2p_k\}$ for $H > 0$.*

**Assumption 3** *i) $F$, $K$ and $\nabla K$ are $L_F$, $L_K$ and $L_{\nabla k}$ Lipschitz; ii) $F$ satisfies the dissipative property, i.e., $\langle F(\boldsymbol{\theta}), \boldsymbol{\theta}\rangle \geq m\|\boldsymbol{\theta}\|^2 - b$ for some $m, b > 0$; iii) Remark 1 applies to the nonconvex setting, i.e. $\sup_{\|f\|_{Lip}\leq 1} |\mathbb{E}_{\boldsymbol{\theta}\sim\mu_\infty}[f(\boldsymbol{\theta})] - \mathbb{E}_{\boldsymbol{\theta}\sim\nu_\infty}[f(\boldsymbol{\theta})]| = \mathcal{W}_1(\rho_\infty, \nu_\infty) = O(M^{-1/2})$.*

**Remark 4** *Assumption 2 is necessary to control the gap between a numerical solution and the exact solution of an SDE. Specifically, it is used to bound the $\tilde{\mathcal{B}}(\hat{\mu}_\infty, \rho_\infty)$ term and the $\tilde{\mathcal{B}}(\mu_T, \hat{\mu}_T)$ term above. Purely relying on the dissipative assumption in Assumption 3 as in non-convex optimization with SG-MCMC (Raginsky et al., 2017; Xu et al., 2018) would induce a bound increasing linearly w.r.t. time $t$. Thus it is not suitable for our goal. Finally, iii) in Assumption 3 is a mild condition and reasonable because we expect particles to be able to approximate all distributions equally well in the asymptotic limit of $t \to \infty$ by ergodicity due to the injected noise. How to remove/replace this assumption is an interesting future work.*

Based on the assumptions above, the bounds for $\tilde{\mathcal{B}}(\hat{\mu}_T, \hat{\mu}_\infty)$ and $\tilde{\mathcal{B}}(\hat{\mu}_\infty, \rho_\infty)$ are summarized below.

**Theorem 9** *Under Assumption 2–3, if we set the stepsize $h_k = h$, we can have the following results:*

$$\tilde{\mathcal{B}}(\hat{\mu}_T, \hat{\mu}_\infty) \leq C_2\varsigma\sigma^{-Md/2}(1 + \varsigma e^{m_{\boldsymbol{\Theta}}h})\exp\left(-2m_{\boldsymbol{\Theta}}Th\sigma^{Md}/\log(\varsigma)\right) \tag{17}$$

$$\tilde{\mathcal{B}}(\hat{\mu}_\infty, \rho_\infty) \leq C_3 h/\beta, \tag{18}$$

*where $\varsigma = 2L_{\boldsymbol{\Theta}}(Mb\beta + m_{\boldsymbol{\Theta}}\beta + Md)/m_{\boldsymbol{\Theta}}$, $L_{\boldsymbol{\Theta}} = \sqrt{2}\beta^{-1}L_F + l'$, $m_{\boldsymbol{\Theta}} = \beta^{-1}m - m'$, and $(\sigma, C_2, C_3, l', m')$ are some positive constants independent of (T, M, h) and $\sigma \in (0,1)$*

**Remark 5** *It is seen that in order to make the $\tilde{\mathcal{B}}(\hat{\mu}_T, \hat{\mu}_\infty)$ term asymptotically decrease to zero, the number of running iteration $T$ should increase at a rate faster enough to compensate the effect of increasing $M$. We believe there is room for improving this bound, which is an interesting future work.*

Next we bound the $\tilde{\mathcal{B}}(\mu_T, \hat{\mu}_T)$ term related to stochastic gradients. By adapting results from linear SDE (Xu et al., 2018), $\tilde{\mathcal{B}}(\mu_T, \hat{\mu}_T)$ can be bounded with Theorem 10.

**Theorem 10** *Under Assumptions 2–3, if we set $B_k = B$ and $h_k = h$, $\tilde{\mathcal{B}}(\mu_T, \hat{\mu}_T)$ is bounded as*

$$\tilde{\mathcal{B}}(\mu_T, \hat{\mu}_T) \leq C_5Th(L_{\boldsymbol{\Theta}}\Gamma' + MC_4)\left((6 + 2\Gamma')\beta/(BM)\right)^{1/2}.$$

*where $\Gamma' = 2(1 + 1/m_{\boldsymbol{\Theta}})(Mb + 2M^2C_4^2 + Md/\beta)$ and , $(C_4, C_5)$ is some positive constant independent of (T, M, h)*

Finally, by combining the results from Theorem 9, 10 and *iii*) in Assumption 3, we arrive at a bound for our target $\tilde{\mathcal{B}}(\mu_T, \nu_\infty)$, summarized in Theorem 11.

**Theorem 11** *Under Assumptions 2–3, there exist some positive constants $(C_2, C_3, C_4, C_5, C_6)$ such that: $\tilde{\mathcal{B}}(\mu_T, \nu_\infty) \leq C_2\varsigma\sigma^{-Md/2}(1 + \varsigma e^{m_{\boldsymbol{\Theta}}h})\exp\left(-2m_{\boldsymbol{\Theta}}Th\sigma^{Md}/\log(\varsigma)\right)$*

$$+ C_3h/\beta + C_5Th(L_{\boldsymbol{\Theta}}\Gamma' + MC_4)\left((6 + 2\Gamma')\beta/(BM)\right)^{1/2} + C_6M^{-1/2},$$

*where $\sigma$, $\varsigma$ and $\Gamma'$ are the same as those in Theorem 9–10.*

## 6 EXPERIMENTS

In this section, we illustrate the effectiveness of our proposed Bayesian sampling framework on several deep-learning models, including Bayesian learning of deep neural network and Bayesian exploration in deep reinforcement learning. We start with a simple illustrative example.

### 6.1 BOUNDS ILLUSTRATION WITH A SIMPLE GAUSSIAN EXAMPLE

We follow Chen et al. (2015) and consider a standard Gaussian model where $x_i \sim \mathcal{N}(\theta, 1), \theta \sim \mathcal{N}(0, 1)$. 1000 data samples $\{x_i\}$ are generated, and every minibatch in the stochastic gradient is of size 10. The test function is defined as $f(\theta) \triangleq \theta^2$, with explicit expression for the posterior average. To evaluate the expectations in the bias and MSE, we average over 200 runs with random initializations. The estimation errors are plotted in Figure 2. It is observed that at the beginning, the errors for the ones with less particles decrease faster than those with more particles. This is reflected in the bounds in

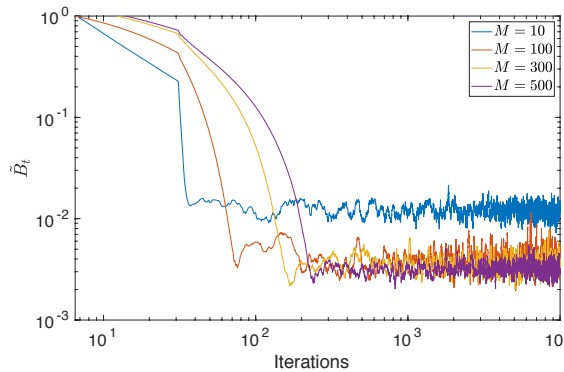

Figure 2: The estimation errors with different particles.

Theorem 15, which are dominated by the bound in Theorem 5 (indicating larger $M$ results in larger errors at the beginning). When more running time/iterations are given, the increase of error in Theorem 5 by increasing $M$ essentially cancels out the exponentially-decay term in Theorem 4. According to Theorem 3, the error would eventually decrease with increasing number of particles.

## 6.2 BAYESIAN NEURAL NETWORKS FOR REGRESSION

We conduct experiments for Bayesian learning of DNNs, where we Bayesian DNNs are used to model weight uncertainty of neural networks, an important topic that has been well explored (Hernández-Lobato & Adams, 2015; Blundell et al., 2015; Li et al., 2016; Louizos & Welling, 2016). We assign simple isotropic Gaussian priors to

Table 1: Averaged predictions with standard deviations in terms of RMSE and log-likelihood on test sets.

| Dataset | Test RMSE | | Test Log likelihood | |
|---|---|---|---|---|
| | SVGD | SPOS | SVGD | SPOS |
| Boston_Housing | $2.961 \pm 0.109$ | $\mathbf{2.829 \pm 0.126}$ | $-2.591 \pm 0.029$ | $\mathbf{-2.532 \pm 0.082}$ |
| Concrete | $6.157 \pm 0.082$ | $\mathbf{5.071 \pm 0.1495}$ | $-3.247 \pm 0.01$ | $\mathbf{-3.062 \pm 0.037}$ |
| Energy | $1.291 \pm 0.029$ | $\mathbf{0.752 \pm 0.0285}$ | $-1.534 \pm 0.026$ | $\mathbf{-1.158 \pm 0.073}$ |
| Kin8nm (0.4) | $0.075 \pm 0.001$ | $\mathbf{0.079 \pm 0.001}$ | $1.138 \pm 0.004$ | $\mathbf{1.092 \pm 0.013}$ |
| Naval (0.4) | $0.004 \pm 0.000$ | $\mathbf{0.004 \pm 0.000}$ | $4.032 \pm 0.008$ | $\mathbf{4.145 \pm 0.02}$ |
| CCPP | $4.127 \pm 0.027$ | $\mathbf{3.939 \pm 0.0495}$ | $-2.843 \pm 0.006$ | $\mathbf{-2.794 \pm 0.025}$ |
| Winequality | $0.604 \pm 0.007$ | $\mathbf{0.598 \pm 0.014}$ | $-0.926 \pm 0.009$ | $\mathbf{-0.911 \pm 0.041}$ |
| Yacht (0.4) | $1.597 \pm 0.099$ | $\mathbf{0.84 \pm 0.0865}$ | $-1.818 \pm 0.06$ | $\mathbf{-1.446 \pm 0.121}$ |
| Protein | $4.392 \pm 0.015$ | $\mathbf{4.254 \pm 0.005}$ | $-2.905 \pm 0.010$ | $\mathbf{-2.876 \pm 0.009}$ |
| YearPredict | $8.684 \pm \mathrm{NA}$ | $\mathbf{8.681 \pm NA}$ | $-3.580 \pm \mathrm{NA}$ | $\mathbf{-3.576 \pm NA}$ |

the weights, and perform posterior sampling with different methods. For all methods, we use a RBF kernel $K(\boldsymbol{\theta}, \boldsymbol{\theta}') = \exp(-\|\boldsymbol{\theta} - \boldsymbol{\theta}'\|_2^2/\eta^2)$, with the bandwidth set to $\eta = \mathrm{med}^2/\log M$. Here med is the median of the pairwise distance between particles. We use a single-layer BNN for regression tasks. Following Li et al. (2015), 10 UCI public datasets are considered: 100 hidden units for 2 large datasets (Protein and YearPredict), and 50 hidden units for the other 8 small datasets. Following Zhang et al. (2018b), we repeat the experiments 20 times with batchsize 100 for all datasets except for Protein and YearPredict, which we repeat 5 times and once with batchsize 1000. The datasets are randomly split into 90% training and 10% testing. For a fair comparison, we use the same split of data (train, val and test) for SVGD and SPOS. The test results are reported on the best model on the validation set. We adopt the root mean squared error (RMSE) and test log-likelihood as the evaluation criteria. The experimental results are shown in Table 1, from where we can see the proposed SPOS outperforms other methods, achieving state-of-the-art results.

## 6.3 BAYESIAN EXPLORATION IN DEEP REINFORCEMENT LEARNING

it is well-accepted that RL performance directly measures how well the uncertainty is learned , reflected by the exploration stage. As a result, we apply our method for RL. Following Liu et al. (2017); Zhang et al. (2018a), we define policies in RL with Bayesian neural networks. This naturally introduces uncertainty into action selections under a specific state-action pair, rendering Bayesian explorations to make policy learning more effective.

Specifically, denote the policy as $\pi_{\boldsymbol{\theta}}(\mathbf{a} \,|\, \mathbf{s})$ parameterized by $\boldsymbol{\theta}$ with prior distribution $p(\boldsymbol{\theta})$, where $\mathbf{a}$ represent the action variable, and $\mathbf{s}$ the state variable. According to Liu et al. (2017), learning the optimal policy corresponds to calculating the following posterior distribution for $\boldsymbol{\theta}$:

$$q(\boldsymbol{\theta}) \propto \exp\left(\frac{1}{\alpha} J(\boldsymbol{\theta})\right) p(\boldsymbol{\theta}), \tag{19}$$

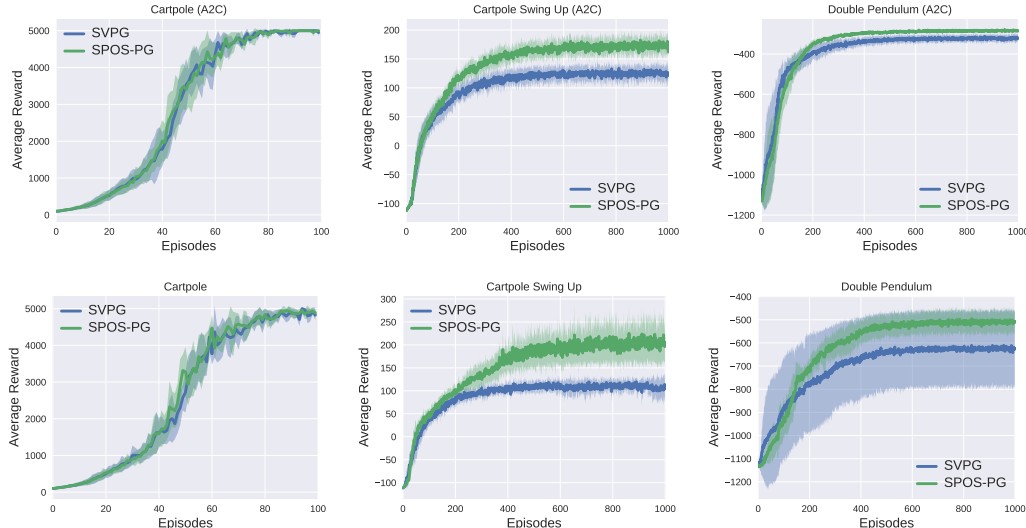

Figure 3: Policy learning with Bayesian exploration in policy-gradient methods on six scenarios with SVPG and SPOS-PG.

where $J(\boldsymbol{\theta})$ denotes the expected cumulative reward under the policy with parameter $\boldsymbol{\theta}$ and $\alpha$ a hyperparameter. Consequently, $\boldsymbol{\theta}$ could be updated by drawing samples from (19) with the proposed SPOS. We denote this method as SPOS-PG. In addition, when drawing samples with SVGD, the resulting algorithm is called Stein variational policy gradient (SVPG) (Liu et al., 2017). Note in implementation, the term $J(\boldsymbol{\theta})$ can be approximated with REINFORCE (Williams, 1992) or advantage actor critic (Schulman et al., 2015), which we will investigate in our experiments.

We follow the same setting as in (Liu et al., 2017) except using simpler policy-network architectures as Houthooft et al. (2016). We conduct experiments on three classical continuous control tasks are considered: Cartpole Swing-Up, Double Pendulum, and Cartpole. Specifically, the policy is parameterized as a two-layer (25-10 hidden units) neural network with `tanh` as the activation function. The maximal length of horizon is set to 500. We use a sample size of 10000 for policy gradient estimation, and $M = 16$, $\alpha = 10$. For the simplest task, Cartpole, all agents are trained for 100 episodes. For the other two complex tasks, all agents are trained up to 1000 episodes. The average reward versus number of episodes are plotted in Figure 3. It is observed that our proposed SPOS-PG obtains much larger average rewards as well as smaller variance compared to SVPG, though the convergence behaviors are similar in the simplest task Carpole.

## 7 CONCLUSION

Motivated by the need of effective and efficient Bayesian sampling techniques in modern deep learning, we propose a probability approach for particle-optimization-based sampling that unifies SG-MCMC and SVGD. Notably, for the first time, by analyzing the corresponding nonlinear SDE, we develop non-asymptotic convergence theory for the proposed SPOS framework, a missing yet important theoretical result since the development of SVGD. Within our theoretical framework, a pitfall of SVGD, which has been studied empirically (Zhuo et al., 2018), is formally analyzed. Our theory also indicates the convergence of SPOS to the true posterior distribution in the asymptotic limit of infinite particles and iterations under appropriate conditions. Our theory is of great practice value, as for the first time it provides nonasymptotic theoretical guarantees for the recently proposed particle-optimization-based algorithms such as the SVGD, whose advantages have also been extensively evaluated by experiments on Bayesian learning of DNNs and Bayesian exploration of DRL. There are a number of interesting future works. For example, one might explore more recently developed techniques such as (Cheng et al., 2018) to improve the convergence bound; one can also adopt the SPOS framework for non-convex optimization like what SG-MCMC is used for, and develop corresponding theory to study the convergence property of the algorithm to the global optimum.

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

## A  DENSITY FUNCTION OF THE MULTI-MODE DISTRIBUTION IN SECTION 3

The negative log-density function of the multi-mode distribution in Section 3 is defined as:

$$U(\boldsymbol{\theta}) \triangleq e^{\frac{3}{4}\boldsymbol{\theta}^2 - \frac{3}{2}\sum_{i=1}^{10} c_i \sin\left(\frac{1}{4}\pi i(\boldsymbol{\theta}+4)\right)} ,$$

where $c = (-0.47, -0.83, -0.71, -0.02, 0.24, 0.01, 0.27, -0.37, 0.87, -0.37)$ is a vector, $c_i$ is the $i$-th element of $c$.

## B  GRONWALL LEMMA

The Gronwall Lemma plays an important role in parts of our proofs, which is stated in Lemma 12.

**Lemma 12 (Gronwall Lemma)** *Let $\mathcal{I}$ denotes an interval of the form $[a, +\infty)$ for some $a \in \mathbb{R}$. If $v(t)$, defined on $\mathcal{I}$, is differentiable in $\mathcal{I}$ and satisfies the following inequality:*

$$v'(t) \leq \beta(t)v(t) ,$$

*where $\beta(t)$ is a real-value continuous function defined on $\mathcal{I}$. Then $v(t)$ can be bounded as:*

$$v(t) \leq v(a) \exp\left(\int_a^t \beta(s)\mathrm{d}s\right)$$

## C  PROOF OF THEOREM 2

To prove Theorem 2, we rely on the definition of generalized derivative in Definition 1.

**Definition 1 (Generalized Derivative)** *Let $g$ and $\phi$ be locally integrable functions on an open set $\Omega \subset \mathbb{R}^d$, that is, Lebesgue integrable on any closed bounded set $\mathcal{F} \subset \omega$. Then $\phi$ is the generalized derivative of $g$ with respect to $\boldsymbol{\theta}$ on $\Omega$, written as $\phi = \partial_{\boldsymbol{\theta}} g$, if for any infinitely-differentiable function $u$ with compact support in $\Omega$, we have*

$$\int_\Omega g(\boldsymbol{\theta})\partial_{\boldsymbol{\theta}} u(\boldsymbol{\theta})\mathrm{d}\boldsymbol{\theta} = -\int_\Omega \phi(\boldsymbol{\theta})u(\boldsymbol{\theta})\mathrm{d}\boldsymbol{\theta} .$$

**Proof** The proof relies on further expansions on the definition of generalized derivative on specific functions. Specifically, let the function $g$ in Definition 1 be in a form of $g \triangleq Gf$ for two functions $G$ and $f$. The generalized derivative of $(Gf)(v, t)$, written as $\partial_{\boldsymbol{\theta}}(Gf)(\boldsymbol{\theta}, t)$, satisfies

$$\int \partial_{\boldsymbol{\theta}}(Gf)(\boldsymbol{\theta}, t)u(\boldsymbol{\theta})\mathrm{d}\boldsymbol{\theta} = -\int Gf(\boldsymbol{\theta}, t)\partial_{\boldsymbol{\theta}} u(\boldsymbol{\theta})\mathrm{d}\boldsymbol{\theta} \tag{20}$$

for all differentiable function $u(\cdot)$.

In Theorem 2, we want to prove a particle representation of the following PDE:

$$\partial_t \mu_t = F_1 = -\nabla_{\boldsymbol{\theta}} \cdot (\mu_t F(\boldsymbol{\theta}_t) + (\mathcal{K} * \mu_t)\mu_t) \triangleq -\partial_{\boldsymbol{\theta}}(Gf)(\boldsymbol{\theta}, t)$$

$$\Rightarrow \int \partial_t \mu_t u(\boldsymbol{\theta})\mathrm{d}\boldsymbol{\theta} = -\int \partial_{\boldsymbol{\theta}}(Gf)(\boldsymbol{\theta}, t)u(\boldsymbol{\theta})\mathrm{d}\boldsymbol{\theta} ,$$

where $f(\boldsymbol{\theta}, t) = \mu_t$. Consequently, we have

$$\int \partial_t f(\boldsymbol{\theta}, t)u(\boldsymbol{\theta})\mathrm{d}\boldsymbol{\theta} = -\int \partial_{\boldsymbol{\theta}}(Gf)(\boldsymbol{\theta}, t)u(\boldsymbol{\theta})\mathrm{d}\boldsymbol{\theta} \tag{21}$$

By applying (20) in (21), we have

$$\int \partial_t f(\boldsymbol{\theta}, t)u(\boldsymbol{\theta})\mathrm{d}\boldsymbol{\theta} = -\int \partial_{\boldsymbol{\theta}}(Gf)(\boldsymbol{\theta}, t)u(\boldsymbol{\theta})\mathrm{d}\boldsymbol{\theta} = \int Gf(\boldsymbol{\theta}, t)\partial_{\boldsymbol{\theta}} u(\boldsymbol{\theta})\mathrm{d}\boldsymbol{\theta} .$$

Since $f \mathrm{d}\boldsymbol{\theta} \triangleq \mu(\mathrm{d}\boldsymbol{\theta}, t)$, we have

$$
\int \partial_t \mu(\mathrm{d}\boldsymbol{\theta}, t) u(\boldsymbol{\theta}) = \int G \cdot \mu(\mathrm{d}\boldsymbol{\theta}, t) \partial_{\boldsymbol{\theta}} u(\boldsymbol{\theta})
$$
$$
\Rightarrow \frac{\mathrm{d}}{\mathrm{d}t} \int \mu(\mathrm{d}\boldsymbol{\theta}, t) u(\boldsymbol{\theta}) = \int G \cdot \mu(\mathrm{d}\boldsymbol{\theta}, t) \partial_{\boldsymbol{\theta}} u(\boldsymbol{\theta})
$$
$$
\Rightarrow \frac{\mathrm{d}}{\mathrm{d}t} \mathbb{E}_{\mu(t)}[u(\boldsymbol{\theta})] = \mathbb{E}_{\mu(t)}[G \cdot \partial_{\boldsymbol{\theta}} u(\boldsymbol{\theta})] . \tag{22}
$$

In particle approximation, we have $\mu(t) = \frac{1}{M} \sum_{i=1}^M \delta_{(\boldsymbol{\theta}_t^{(i)})}(\boldsymbol{\theta})$. For each particle, let $u(\boldsymbol{\theta}) = \boldsymbol{\theta}$, (22) reduces to the following equation:

$$
\mathrm{d}\boldsymbol{\theta}_t^{(i)} = G(\boldsymbol{\theta}_t^{(i)}) \mathrm{d}t .
$$

This completes the proof. ∎

## D  PROOF OF THEOREM 3

Note that one challenge in our analysis compared with the analysis for linear SDEs, such as those for SG-MCMC Vollmer et al. (2016); Chen et al. (2015), is how to bound the gap between the original nonlinear PDE (4) and the reduced nonlinear SDE (8). Based on the techniques on analyzing granular media equations in Malrieu (2003); Cattiaux et al. (2008); Durmus et al. (2018), we introduce a nonlinear SDE as an element in-between (6) and (8) like :

$$
\begin{cases}
\mathrm{d}\bar{\boldsymbol{\theta}}_t = -\beta^{-1} F(\bar{\boldsymbol{\theta}}_t) - E_{Y \sim \nu_t} K(\bar{\boldsymbol{\theta}}_t - Y) F(Y) + \nabla K * \nu_t(\bar{\boldsymbol{\theta}}_t) + \sqrt{2\beta^{-1}} \mathrm{d}\bar{\mathcal{W}}_t \\
\mathcal{L}(\bar{\boldsymbol{\theta}}_t) = \nu_t \mathrm{d}\boldsymbol{\theta}
\end{cases} \tag{23}
$$

where $\mathcal{L}(\bar{\boldsymbol{\theta}}_t)$ denotes the probability law of $\bar{\boldsymbol{\theta}}_t$, $\bar{\mathcal{W}}_t \in \mathbb{R}^d$ is a $d$-dimensional Brownian motion and $Y$ is a random variable independent of $\bar{\boldsymbol{\theta}}_t$ and just used here for the sake of clarity. In order to match the SDE system (8) of the particles $\{\boldsymbol{\theta}_t^{(i)}\}_{i=1}^M$, we duplicate (23) $M$ times, each endowing with an exact solution $\bar{\boldsymbol{\theta}}_t^{(i)}$ indexed by $i$. The distribution of each particles $\{\bar{\boldsymbol{\theta}}_t^{(i)}\}_{i=1}^M$ is $\nu_t$ and the corresponding $\bar{\mathcal{W}}_t^{(i)}$ can be set exactly the same as the $\mathcal{W}_t^{(i)}$ in (8):

$$
\begin{cases}
\mathrm{d}\bar{\boldsymbol{\theta}}_t^{(i)} = -\beta^{-1} F(\bar{\boldsymbol{\theta}}_t^{(i)}) - E_{Y_i \sim \nu_t} K(\bar{\boldsymbol{\theta}}_t^{(i)} - Y_i) F(Y_i) + \nabla K * \nu_t(\bar{\boldsymbol{\theta}}_t^{(i)}) + \sqrt{2\beta^{-1}} \mathrm{d}\bar{\mathcal{W}}_t^{(i)} \\
\mathcal{L}(\bar{\boldsymbol{\theta}}_t^{(i)}) = \nu_t \mathrm{d}\boldsymbol{\theta}
\end{cases}
$$
$$
\tag{24}
$$

where $Y_i$ is a random variable independent of $\bar{\boldsymbol{\theta}}_t^{(i)}$ and just used here for the convenience of the proof.

**Proof** [Proof of Theorem 3] Firstly we have

$$
\mathrm{d}\left(\boldsymbol{\theta}_t^{(i)} - \bar{\boldsymbol{\theta}}_t^{(i)}\right) = -\beta^{-1}\left(F(\boldsymbol{\theta}_t^{(i)}) - F(\bar{\boldsymbol{\theta}}_t^{(i)})\right)\mathrm{d}t + \frac{1}{M}\sum_j^M \left[\nabla K(\boldsymbol{\theta}_t^{(i)} - \boldsymbol{\theta}_t^{(j)}) - \nabla K * \nu_t(\bar{\boldsymbol{\theta}}_t^{(i)})\right]\mathrm{d}t
$$
$$
- \frac{1}{M}\sum_j^M \left(F(\boldsymbol{\theta}_t^{(j)})W(\boldsymbol{\theta}_t^{(i)} - \boldsymbol{\theta}_t^{(j)}) - \mathbb{E}_{Y_j \sim \nu_t} F(Y_j)W(\bar{\boldsymbol{\theta}}_t^{(i)} - Y_j)\right)\mathrm{d}t
$$

$$
\Rightarrow \mathrm{d}\left(\sum_i^M \left\|\boldsymbol{\theta}_t^{(i)} - \bar{\boldsymbol{\theta}}_t^{(i)}\right\|^2\right) = \frac{2}{M}\sum_{i,j}^M (A_{ij}(t) + B_{ij}(t) + C_{ij}(t) + F_{ij}(t) + G_{ij}(t) + H_{ij}(t))\mathrm{d}t
$$
$$
\tag{25}
$$

where $A_{ij}(t) = -\beta^{-1}\left(F(\boldsymbol{\theta}_t^{(i)}) - F(\bar{\boldsymbol{\theta}}_t^{(i)})\right) \cdot \left(\boldsymbol{\theta}_t^{(i)} - \bar{\boldsymbol{\theta}}_t^{(i)}\right)$

$\quad B_{ij}(t) = \left(\nabla K(\boldsymbol{\theta}_t^{(i)} - \boldsymbol{\theta}_t^{(j)}) - \nabla K(\bar{\boldsymbol{\theta}}_t^{(i)} - \bar{\boldsymbol{\theta}}_t^{(j)})\right) \cdot \left(\boldsymbol{\theta}_t^{(i)} - \bar{\boldsymbol{\theta}}_t^{(i)}\right)$

$$C_{ij}(t) = \left(\nabla K(\bar{\boldsymbol{\theta}}_t^{(i)} - \bar{\boldsymbol{\theta}}_t^{(j)}) - \nabla K * \nu_t(\bar{\boldsymbol{\theta}}_t^{(i)})\right) \cdot \left(\boldsymbol{\theta}_t^{(i)} - \bar{\boldsymbol{\theta}}_t^{(i)}\right)$$

$$F_{ij}(t) = -\left(F(\boldsymbol{\theta}_t^{(j)})K(\boldsymbol{\theta}_t^{(i)} - \boldsymbol{\theta}_t^{(j)}) - F(\bar{\boldsymbol{\theta}}_t^{(j)})K(\boldsymbol{\theta}_t^{(i)} - \boldsymbol{\theta}_t^{(j)})\right) \cdot \left(\boldsymbol{\theta}_t^{(i)} - \bar{\boldsymbol{\theta}}_t^{(i)}\right)$$

$$G_{ij}(t) = -\left(F(\bar{\boldsymbol{\theta}}_t^{(j)})K(\boldsymbol{\theta}_t^{(i)} - \boldsymbol{\theta}_t^{(j)}) - F(\bar{\boldsymbol{\theta}}_t^{(j)})K(\bar{\boldsymbol{\theta}}_t^{(i)} - \bar{\boldsymbol{\theta}}_t^{(j)})\right) \cdot \left(\boldsymbol{\theta}_t^{(i)} - \bar{\boldsymbol{\theta}}_t^{(i)}\right)$$

$$H_{ij}(t) = -\left(F(\bar{\boldsymbol{\theta}}_t^{(j)})K(\bar{\boldsymbol{\theta}}_t^{(i)} - \bar{\boldsymbol{\theta}}_t^{(j)}) - \mathbb{E}_{Y_j \sim \nu_t}F(Y_j)K(\bar{\boldsymbol{\theta}}_t^{(i)} - Y_j)\right) \cdot \left(\boldsymbol{\theta}_t^{(i)} - \bar{\boldsymbol{\theta}}_t^{(i)}\right)$$

For the $A_{ij}(t)$ term, according to the $i$) in Assumption 1 for $F$, we have

$$\sum_{ij} A_{ij}(t) = -\sum_{ij} \beta^{-1}\left(F(\boldsymbol{\theta}_t^{(i)}) - F(\bar{\boldsymbol{\theta}}_t^{(i)})\right) \cdot \left(\boldsymbol{\theta}_t^{(i)} - \bar{\boldsymbol{\theta}}_t^{(i)}\right)$$

$$\leq -\beta^{-1}m_F M \sum_i \left\|\boldsymbol{\theta}_t^{(i)} - \bar{\boldsymbol{\theta}}_t^{(i)}\right\|^2$$

For the $B_{ij}(t)$ term, applying the concave condition for $K$ and the oddness of $\nabla K$ in Assumption 1, we have

$$\sum_{ij} B_{ij}(t) = \sum_{ij}^{M} \left(\nabla K(\boldsymbol{\theta}_t^{(i)} - \boldsymbol{\theta}_t^{(j)}) - \nabla K(\bar{\boldsymbol{\theta}}_t^{(i)} - \bar{\boldsymbol{\theta}}_t^{(j)})\right) \cdot \left(\boldsymbol{\theta}_t^{(i)} - \bar{\boldsymbol{\theta}}_t^{(i)}\right)$$

$$= \frac{1}{2}\sum_{ij}^{M} \left(\nabla K(\boldsymbol{\theta}_t^{(i)} - \boldsymbol{\theta}_t^{(j)}) - \nabla K(\bar{\boldsymbol{\theta}}_t^{(i)} - \bar{\boldsymbol{\theta}}_t^{(j)})\right) \cdot \left(\boldsymbol{\theta}_t^{(i)} - \bar{\boldsymbol{\theta}}_t^{(i)} - (\boldsymbol{\theta}_t^{(j)} - \bar{\boldsymbol{\theta}}_t^{(j)})\right)$$

$$\leq -\frac{1}{2}m_K \sum_{ij}^{M} \left\|\boldsymbol{\theta}_t^{(i)} - \bar{\boldsymbol{\theta}}_t^{(i)} - (\boldsymbol{\theta}_t^{(j)} - \bar{\boldsymbol{\theta}}_t^{(j)})\right\|^2 \leq 0$$

For the $C_{ij}(t)$ term, we have

$$\mathbb{E}\sum_j C_{ij}(t) \overset{(1)}{\leq} \left(\mathbb{E}\left\|\boldsymbol{\theta}_t^{(i)} - \bar{\boldsymbol{\theta}}_t^{(i)}\right\|^2\right)^{1/2} \left(\mathbb{E}\left\|\sum_j \left(\nabla K(\bar{\boldsymbol{\theta}}_t^{(i)} - \bar{\boldsymbol{\theta}}_t^{(j)}) - \nabla K * \nu_t(\bar{\boldsymbol{\theta}}_t^{(i)})\right)\right\|^2\right)^{1/2}$$

$$\overset{(2)}{=} \left(\mathbb{E}\left\|\boldsymbol{\theta}_t^{(i)} - \bar{\boldsymbol{\theta}}_t^{(i)}\right\|^2\right)^{1/2} \left(\sum_j \mathbb{E}\left(\nabla K(\bar{\boldsymbol{\theta}}_t^{(i)} - \bar{\boldsymbol{\theta}}_t^{(j)}) - \nabla K * \nu_t(\bar{\boldsymbol{\theta}}_t^{(i)})\right)^2\right)^{1/2}$$

$$\overset{(3)}{\leq} H_{\nabla K}\sqrt{2M}\left(\mathbb{E}\left\|\boldsymbol{\theta}_t^{(i)} - \bar{\boldsymbol{\theta}}_t^{(i)}\right\|^2\right)^{1/2}$$

where the (1) is obtained by applying the Cauchy-Schwarz inequality and (2) by the fact that $\mathbb{E}\left(K(\bar{\boldsymbol{\theta}}_t^{(i)} - \bar{\boldsymbol{\theta}}_t^{(j)}) - K * \nu_t(\bar{\boldsymbol{\theta}}_t^{(i)})\right) = 0$. We can tune the bandwidth of the RBF kernel to make $\|\nabla K\| \leq H_{\nabla K}$. Hence (3) is obtain by the boundedness of $\nabla K(\boldsymbol{\theta})$.

Similarly, since $K \leq 1$, we have the following result for $H_{ij}(t)$ term,

$$\mathbb{E}\sum_j H_{ij}(t)$$

$$\leq \left(\mathbb{E}\left\|\boldsymbol{\theta}_t^{(i)} - \bar{\boldsymbol{\theta}}_t^{(i)}\right\|^2\right)^{1/2} \left(\mathbb{E}\left\|\sum_j \left(F(\bar{\boldsymbol{\theta}}_t^{(j)})K(\bar{\boldsymbol{\theta}}_t^{(i)} - \bar{\boldsymbol{\theta}}_t^{(j)}) - \mathbb{E}_{Y_j \sim \nu_t}F(Y_j)K(\bar{\boldsymbol{\theta}}_t^{(i)} - Y_j)\right)\right\|^2\right)^{1/2}$$

$$= \left(\mathbb{E}\left\|\boldsymbol{\theta}_t^{(i)} - \bar{\boldsymbol{\theta}}_t^{(i)}\right\|^2\right)^{1/2} \left(\sum_j \mathbb{E}\left(F(\bar{\boldsymbol{\theta}}_t^{(j)})K(\bar{\boldsymbol{\theta}}_t^{(i)} - \bar{\boldsymbol{\theta}}_t^{(j)}) - \mathbb{E}_{Y_j \sim \nu_t}F(Y_j)K(\bar{\boldsymbol{\theta}}_t^{(i)} - Y_j)\right)^2\right)^{1/2}$$

$$\leq H_F \sqrt{2M} \left( \mathbb{E} \left\| \boldsymbol{\theta}_t^{(i)} - \bar{\boldsymbol{\theta}}_t^{(i)} \right\|^2 \right)^{1/2}$$

For the $F_{ij}(t)$ and $G_{ij}(t)$ terms, we have:

$$\sum_{ij} F_{ij}(t) = -\sum_{ij} \left( F(\boldsymbol{\theta}_t^{(j)}) K(\boldsymbol{\theta}_t^{(i)} - \boldsymbol{\theta}_t^{(j)}) - F(\bar{\boldsymbol{\theta}}_t^{(j)}) K(\boldsymbol{\theta}_t^{(i)} - \boldsymbol{\theta}_t^{(j)}) \right) \cdot \left( \boldsymbol{\theta}_t^{(i)} - \bar{\boldsymbol{\theta}}_t^{(i)} \right)$$

$$\leq \sum_{ij} L_F \left\| \boldsymbol{\theta}_t^{(j)} - \bar{\boldsymbol{\theta}}_t^{(j)} \right\| \left\| \boldsymbol{\theta}_t^{(i)} - \bar{\boldsymbol{\theta}}_t^{(i)} \right\|$$

$$\leq 2 L_F M \sum_i \left\| \boldsymbol{\theta}_t^{(i)} - \bar{\boldsymbol{\theta}}_t^{(i)} \right\|^2 .$$

$$\sum_{ij} G_{ij}(t) = \sum_{ij} \left( F(\bar{\boldsymbol{\theta}}_t^{(j)}) K(\boldsymbol{\theta}_t^{(i)} - \boldsymbol{\theta}_t^{(j)}) - F(\bar{\boldsymbol{\theta}}_t^{(j)}) K(\bar{\boldsymbol{\theta}}_t^{(i)} - \bar{\boldsymbol{\theta}}_t^{(j)}) \right) \cdot \left( \boldsymbol{\theta}_t^{(i)} - \bar{\boldsymbol{\theta}}_t^{(i)} \right)$$

$$\leq H_F L_K \sum_{ij} \left\| \boldsymbol{\theta}_t^{(i)} - \bar{\boldsymbol{\theta}}_t^{(i)} - (\boldsymbol{\theta}_t^{(j)} - \bar{\boldsymbol{\theta}}_t^{(j)}) \right\| \left\| \boldsymbol{\theta}_t^{(i)} - \bar{\boldsymbol{\theta}}_t^{(i)} \right\|$$

$$\leq 3 H_F L_K M \sum_i \left\| \boldsymbol{\theta}_t^{(i)} - \bar{\boldsymbol{\theta}}_t^{(i)} \right\|^2$$

We denote $\gamma_i(t) \triangleq \mathbb{E} \left\| \boldsymbol{\theta}_t^{(i)} - \bar{\boldsymbol{\theta}}_t^{(i)} \right\|^2$. Due to the exchangeability of the particles, $\gamma_i(t)$ are the same for all the particles, denoted as $\gamma(t)$. Then according to (25), we have

$$\gamma'(t) \leq -2\lambda_1 \gamma(t) + \frac{H_{\nabla K}\sqrt{2} + H_F \sqrt{2}}{\sqrt{M}} \sqrt{\gamma(t)} .$$
$$\text{where } \lambda_1 = \beta^{-1} m_F - 3 H_F L_K - 2 L_F$$

$$\Rightarrow (\sqrt{\gamma(t)} - \frac{(H_{\nabla K} + H_F)/\sqrt{2}}{\sqrt{M}(\beta^{-1} - 3 H_F L_K - 2 L_F)})' \leq -\lambda_1 (\sqrt{\gamma(t)} - \frac{(H_{\nabla K} + H_F)/\sqrt{2}}{\sqrt{M}(\beta^{-1} - 3 H_F L_K - 2 L_F)})$$

Note that $\boldsymbol{\theta}_t^{(i)}$ and $\bar{\boldsymbol{\theta}}_t^{(i)}$ are initialized with the same initial distribution $\mu_0 = \nu_0$. In the definition of $\bar{\boldsymbol{\theta}}_t^{(i)}$, there is no restriction on how the initial value is set. As a result, we can set $\boldsymbol{\theta}_0^{(i)}$ to be identical to $\bar{\boldsymbol{\theta}}_0^{(i)}$, leading to $\gamma(0) = 0$. Then according to the Gronwall Lemma, we have

$$\sqrt{\gamma(t)} \leq \frac{(H_{\nabla K} + H_F)/\sqrt{2}}{\sqrt{M}(\beta^{-1} - 3 H_F L_K - 2 L_F)}$$

Hence, there exist some positive constant $(c_1, c_2)$ such that:

$$W_1(\rho_t, \nu_t) \overset{(1)}{\leq} W_2(\rho_t, \nu_t)$$
$$\overset{(2)}{\leq} \sqrt{\mathbb{E} \left\| \boldsymbol{\theta}_t^{(i)} - \bar{\boldsymbol{\theta}}_t^{(i)} \right\|^2} \overset{(3)}{\leq} \frac{c_1}{\sqrt{M}(\beta^{-1} - c_2)}, \tag{26}$$

where (1) holds due to the relationship between $W_1$ and $W_2$ metric Givens & Shortt (1984), (2) due to the definition of $W_2$ and (3) due to the result from the previous proof.

∎

# E  PROOF OF THEOREM 4

**Proof** [Proof of Theorem 4] Firstly, what we aim at is $W_1(\nu_t, \nu_\infty) \leq c_3 \exp(-2\lambda_1 t)$ in this theorem. According to the relationship between $W_1$ and $W_2$ metric Givens & Shortt (1984), once we bound $W_2(\nu_t, \nu_\infty)$ as $W_2(\nu_t, \nu_\infty) \leq c_3 \exp(-2\lambda_1 t)$, we will finish our proof.

Next, look at the equation (8):

If we set the initial distribution of each particle to be $\nu_0$, which means $\rho_0 = \mathcal{L}(\boldsymbol{\theta}_0^{(i)}) = \nu_0$, we will derive M particles denoted as $\{\boldsymbol{\theta}_{t,1}^{(i)}\}_{i=1}^M$. We denote the distribution of each $\boldsymbol{\theta}_{t,1}^{(i)}$ at $t$ as $\rho_{t,1}$.

If we set the initial distribution of each particle to be $\nu_\infty$, which means $\rho_0 = \mathcal{L}(\boldsymbol{\theta}_0^{(i)}) = \nu_\infty$, we will derive M particles denoted as $\{\boldsymbol{\theta}_{t,2}^{(i)}\}_{i=1}^M$. We denote the distribution of each $\boldsymbol{\theta}_{t,2}^{(i)}$ at $t$ as $\rho_{t,2}$.

Since we need to bound $W_2(\nu_t, \nu_\infty)$, we make the following decomposition:

$$W_2(\nu_t, \nu_\infty) \leq W_2(\nu_t, \rho_{t,1}) + W_2(\rho_{t,1}, \rho_{t,2}) + W_2(\rho_{t,2}, \nu_\infty). \tag{27}$$

Note that $\rho_{0,1} = \nu_0$ and $\rho_{0,2} = \nu_\infty$. Then, according to (26), we have

$$W_2(\nu_t, \rho_{t,1}) \leq \frac{c_1}{\sqrt{M}(\beta^{-1} - c_2)} \quad and \quad W_2(\rho_{t,2}, \nu_\infty) \leq \frac{c_1}{\sqrt{M}(\beta^{-1} - c_2)}$$

Now we need to focus on the term $W_2(\rho_{t,1}, \rho_{t,2})$. Since $W_2(\rho_{t,1}, \rho_{t,2}) \leq \mathbb{E}\left(\left\|\boldsymbol{\theta}_{t,1}^{(i)} - \boldsymbol{\theta}_{t,2}^{(i)}\right\|^2\right) \triangleq r(t)$, we will derive a bound for $\mathbb{E}\left(\left\|\boldsymbol{\theta}_{t,1}^{(i)} - \boldsymbol{\theta}_{t,2}^{(i)}\right\|^2\right)$ in the following. We have

$$d\left(\boldsymbol{\theta}_{t,1}^{(i)} - \boldsymbol{\theta}_{t,2}^{(i)}\right) = -\beta^{-1}\left(F(\boldsymbol{\theta}_{t,1}^{(i)}) - F(\boldsymbol{\theta}_{t,2}^{(i)})\right)dt$$

$$+ \frac{1}{M}\sum_j^M\left[\nabla K(\boldsymbol{\theta}_{t,1}^{(i)} - \boldsymbol{\theta}_{t,1}^{(j)}) - \nabla K(\boldsymbol{\theta}_{t,2}^{(i)} - \boldsymbol{\theta}_{t,2}^{(j)})\right]dt$$

$$- \frac{1}{M}\sum_j^M\left(F(\boldsymbol{\theta}_{t,1}^{(j)})K(\boldsymbol{\theta}_{t,1}^{(i)} - \boldsymbol{\theta}_{t,1}^{(j)}) - F(\boldsymbol{\theta}_{t,2}^{(j)})K(\boldsymbol{\theta}_{t,2}^{(i)} - \boldsymbol{\theta}_{t,2}^{(j)})\right)dt$$

$$\Rightarrow d\left(\sum_i^M\left\|\boldsymbol{\theta}_{t,1}^{(i)} - \boldsymbol{\theta}_{t,2}^{(i)}\right\|^2\right) = \frac{2}{M}\sum_{i,j}^M(\xi_{ij}^1(t) + \xi_{ij}^2(t) + \xi_{ij}^3(t) + \xi_{ij}^4(t))dt$$

where $\xi_{ij}^1(t) = -\beta^{-1}\left(F(\boldsymbol{\theta}_{t,1}^{(i)}) - F(\boldsymbol{\theta}_{t,2}^{(i)})\right) \cdot \left(\boldsymbol{\theta}_{t,1}^{(i)} - \boldsymbol{\theta}_{t,2}^{(i)}\right)$

$\xi_{ij}^2(t) = \left(\nabla K(\boldsymbol{\theta}_{t,1}^{(i)} - \boldsymbol{\theta}_{t,1}^{(j)}) - \nabla K(\boldsymbol{\theta}_{t,2}^{(i)} - \boldsymbol{\theta}_{t,2}^{(j)})\right) \cdot \left(\boldsymbol{\theta}_{t,1}^{(i)} - \boldsymbol{\theta}_{t,2}^{(i)}\right)$

$\xi_{ij}^3(t) = -\left(F(\boldsymbol{\theta}_{t,1}^{(j)})K(\boldsymbol{\theta}_{t,1}^{(i)} - \boldsymbol{\theta}_{t,1}^{(j)}) - F(\boldsymbol{\theta}_{t,2}^{(j)})K(\boldsymbol{\theta}_{t,1}^{(i)} - \boldsymbol{\theta}_{t,1}^{(j)})\right) \cdot \left(\boldsymbol{\theta}_{t,1}^{(i)} - \boldsymbol{\theta}_{t,2}^{(i)}\right)$

$\xi_{ij}^4(t) = -\left(F(\boldsymbol{\theta}_{t,2}^{(j)})K(\boldsymbol{\theta}_{t,1}^{(i)} - \boldsymbol{\theta}_{t,1}^{(j)}) - F(\boldsymbol{\theta}_{t,2}^{(j)})K(\boldsymbol{\theta}_{t,2}^{(i)} - \boldsymbol{\theta}_{t,2}^{(j)})\right) \cdot \left(\boldsymbol{\theta}_{t,1}^{(i)} - \boldsymbol{\theta}_{t,2}^{(i)}\right)$

For the $\xi_{ij}^1(t)$ terms, according to the $i)$ in Assumption 1 for $F$, we have

$$\sum_{ij}\xi_{ij}^1(t) = -\sum_{ij}\beta^{-1}\left(F(\boldsymbol{\theta}_{t,1}^{(i)}) - F(\boldsymbol{\theta}_{t,2}^{(i)})\right) \cdot \left(\boldsymbol{\theta}_{t,1}^{(i)} - \boldsymbol{\theta}_{t,2}^{(i)}\right)$$

$$\leq -\beta^{-1}m_F M\sum_i\left\|\boldsymbol{\theta}_{t,1}^{(i)} - \boldsymbol{\theta}_{t,2}^{(i)}\right\|^2.$$

For the $\xi_{ij}^2(t)$ term, applying the concave condition for $K$ and the oddness of $\nabla K$ in Assumption 1, we have

$$\sum_{ij} \xi_{ij}^2(t) = \sum_{ij}^{M} \left( \nabla K(\boldsymbol{\theta}_{t,1}^{(i)} - \boldsymbol{\theta}_{t,1}^{(j)}) - \nabla K(\boldsymbol{\theta}_{t,2}^{(i)} - \boldsymbol{\theta}_{t,2}^{(j)}) \right) \cdot \left( \boldsymbol{\theta}_{t,1}^{(i)} - \boldsymbol{\theta}_{t,2}^{(i)} \right)$$

$$= \frac{1}{2} \sum_{ij}^{M} \left( \nabla K(\boldsymbol{\theta}_{t,1}^{(i)} - \boldsymbol{\theta}_{t,1}^{(j)}) - \nabla K(\boldsymbol{\theta}_{t,2}^{(i)} - \boldsymbol{\theta}_{t,2}^{(j)}) \right) \cdot \left( \boldsymbol{\theta}_{t,1}^{(i)} - \boldsymbol{\theta}_{t,2}^{(i)} - (\boldsymbol{\theta}_{t,1}^{(j)} - \boldsymbol{\theta}_{t,2}^{(j)}) \right)$$

$$\leq -\frac{1}{2} m_K \sum_{ij}^{M} \left\| \boldsymbol{\theta}_{t,1}^{(i)} - \boldsymbol{\theta}_{t,2}^{(i)} - (\boldsymbol{\theta}_{t,1}^{(j)} - \boldsymbol{\theta}_{t,2}^{(j)}) \right\|^2 \leq 0 .$$

For the $\xi_{ij}^3(t)$ terms, after applying the $L_F$-Lipschitz property for $F$ and $K \leq 1$, we have

$$\sum_{ij} \xi_{ij}^3(t) = \sum_{ij} - \left( F(\boldsymbol{\theta}_{t,1}^{(j)}) K(\boldsymbol{\theta}_{t,1}^{(i)} - \boldsymbol{\theta}_{t,1}^{(j)}) - F(\boldsymbol{\theta}_{t,2}^{(j)}) K(\boldsymbol{\theta}_{t,1}^{(i)} - \boldsymbol{\theta}_{t,1}^{(j)}) \right) \cdot \left( \boldsymbol{\theta}_{t,1}^{(i)} - \boldsymbol{\theta}_{t,2}^{(i)} \right)$$

$$\leq \sum_{ij} L_F \left\| \boldsymbol{\theta}_{t,1}^{(j)} - \boldsymbol{\theta}_{t,2}^{(j)} \right\| \left\| \boldsymbol{\theta}_{t,1}^{(i)} - \boldsymbol{\theta}_{t,2}^{(i)} \right\|$$

$$\leq 2 L_F M \sum_i \left\| \boldsymbol{\theta}_{t,1}^{(i)} - \boldsymbol{\theta}_{t,2}^{(i)} \right\|^2 .$$

For the $\xi_{ij}^4(t)$ terms :

$$\sum_{ij} \xi_{ij}^4(t) = -\sum_{ij} \left( F(\boldsymbol{\theta}_{t,2}^{(j)}) K(\boldsymbol{\theta}_{t,1}^{(i)} - \boldsymbol{\theta}_{t,1}^{(j)}) - F(\boldsymbol{\theta}_{t,2}^{(j)}) K(\boldsymbol{\theta}_{t,2}^{(i)} - \boldsymbol{\theta}_{t,2}^{(j)}) \right) \cdot \left( \boldsymbol{\theta}_{t,1}^{(i)} - \boldsymbol{\theta}_{t,2}^{(i)} \right)$$

$$\leq H_F L_K \sum_{ij} \left\| \boldsymbol{\theta}_{t,1}^{(i)} - \boldsymbol{\theta}_{t,2}^{(i)} - (\boldsymbol{\theta}_{t,1}^{(j)} - \boldsymbol{\theta}_{t,2}^{(j)}) \right\| \left\| \boldsymbol{\theta}_{t,1}^{(i)} - \boldsymbol{\theta}_{t,2}^{(i)} \right\|$$

$$\leq 3 H_F L_K M \sum_i \left\| \boldsymbol{\theta}_{t,1}^{(i)} - \boldsymbol{\theta}_{t,2}^{(i)} \right\|^2 .$$

Now we have

$$r'(t) \leq -2(\beta^{-1} m_F - 3 H_F L_K - 2 L_F) r(t) .$$

According to the Gronwall lemma,

$$r(t) \leq r(0) e^{-2\lambda_1 t},$$

where $\lambda_1 = \beta^{-1} m_F - 3 H_F L_W - 2 L_F$.

Consequently, there exists some positive constant $c_3$ such that

$$W_2(\rho_{t,1}, \rho_{t,2}) \leq c_3 e^{-2\lambda_1 t}$$

Then we have

$$W_2(\nu_t, \nu_\infty) \leq c_3 e^{-2\lambda_1 t} + \frac{c_1}{\sqrt{M}(\beta^{-1} - c_2)} + \frac{c_1}{\sqrt{M}(\beta^{-1} - c_2)}$$

However, it worth noting that $\nu_t$ is the solution of (6) which has nothing to do with the number of particles, $M$. Then let $M \to \infty$, we can derive that $W_2(\nu_t, \nu_\infty) \leq c_3 e^{-2\lambda_1 t}$. Now we finish our proof. ∎

# F  PROOF OF THEOREM 5

To bound the $W_1(\mu_T, \rho_{\sum_{k=0}^{T-1} h_k})$ term, note the original SDE driving the particles $\{\boldsymbol{\theta}_t^{(i)}\}$ in (8) is a nonlinear SDE, which is hard to deal with. Fortunately, (8) can be turned into a linear SDE by concatenating the particles at each time into a single vector representation, *i.e.*, by defining the new parameter at time $t$ as $\boldsymbol{\Theta}_t \triangleq [\boldsymbol{\theta}_t^{(1)}, \cdots, \boldsymbol{\theta}_t^{(M)}] \in \mathbb{R}^{Md}$. Consequently, $\boldsymbol{\Theta}_t$ is driven by the following linear SDE:

$$\mathrm{d}\boldsymbol{\Theta}_t = -F_{\boldsymbol{\Theta}}(\boldsymbol{\Theta}_t)\mathrm{d}t + \sqrt{2\beta^{-1}}\mathrm{d}\mathcal{W}_t^{(Md)} , \tag{28}$$

where $F_{\boldsymbol{\Theta}}(\boldsymbol{\Theta}_t) \triangleq [\beta^{-1}F(\boldsymbol{\theta}_t^{(1)}) - \frac{1}{M}\sum_{j=1}^M \nabla K(\boldsymbol{\theta}_t^{(1)} - \boldsymbol{\theta}_t^{(j)}) + \frac{1}{M}\sum_{j=1}^M K(\boldsymbol{\theta}_t^{(1)} - \boldsymbol{\theta}_t^{(j)})F(\boldsymbol{\theta}_t^{(j)}), \cdots, \beta^{-1}F(\boldsymbol{\theta}_t^{(M)}) - \frac{1}{M}\sum_{j=1}^M \nabla K(\boldsymbol{\theta}_t^{(M)} - \boldsymbol{\theta}_t^{(j)}) + \frac{1}{M}\sum_{j=1}^M K(\boldsymbol{\theta}_t^{(M)} - \boldsymbol{\theta}_t^{(j)})F(\boldsymbol{\theta}_t^{(j)})]$ is a vector function $\mathbb{R}^{Md} \to \mathbb{R}^{Md}$, and $\mathcal{W}_t^{(Md)}$ is Brownian motion of dimension $Md$.

Now we define the $F_{(q)\boldsymbol{\Theta}}(\boldsymbol{\Theta}_t) \triangleq [\beta^{-1}F_q(\boldsymbol{\theta}_t^{(1)}) - \frac{1}{MN}\sum_{j=1}^M \nabla K(\boldsymbol{\theta}_t^{(1)} - \boldsymbol{\theta}_t^{(j)}) + \frac{1}{M}\sum_{j=1}^M K(\boldsymbol{\theta}_t^{(1)} - \boldsymbol{\theta}_t^{(j)})F_q(\boldsymbol{\theta}_t^{(j)}), \cdots, \beta^{-1}F_q(\boldsymbol{\theta}_t^{(M)}) - \frac{1}{MN}\sum_{j=1}^M \nabla K(\boldsymbol{\theta}_t^{(M)} - \boldsymbol{\theta}_t^{(j)}) + \frac{1}{M}\sum_{j=1}^M K(\boldsymbol{\theta}_t^{(M)} - \boldsymbol{\theta}_t^{(j)})F_q(\boldsymbol{\theta}_t^{(j)})]$. We can verify that $F_{\boldsymbol{\Theta}}(\boldsymbol{\Theta}_t) = \sum_{q=1}^N F_{(q)\boldsymbol{\Theta}}(\boldsymbol{\Theta}_t)$.

Then we define $\Theta_k \triangleq [\theta_k^{(1)}, \cdots, \theta_k^{(M)}]$ and $G_{\mathcal{I}_k}^{\Theta} \triangleq \frac{N}{B_k}\sum_{q \in \mathcal{I}_k} F_{(q)\boldsymbol{\Theta}}(\Theta_k)$. We can verify that the following result holds:

$$\Theta_{k+1} = \Theta_k - \beta^{-1}G_{\mathcal{I}_k}^{\Theta}h_k + \sqrt{2\beta^{-1}h_k}\Xi_k , \tag{29}$$

where $\Xi_k \sim \mathcal{N}(\mathbf{0}, \mathbf{I}_{Md \times Md})$. Now we reach the conclusion that $\Theta_k$ of (29) is accutually the numerical solution of the SDE (28) via stochastic gradients.

We denote the distribution of $\Theta_k$ as $\mu_k^{\Theta}$ and the distribution of $\boldsymbol{\Theta}_t$ as $\rho_t^{\boldsymbol{\Theta}}$. Before proceeding to our theoretical results, we need to present the following Lemmas which is very important in our proof.

**Lemma 13** $W_1(\mu_k, \rho_t) \leq \frac{1}{\sqrt{M}}W_1(\mu_k^{\Theta}, \rho_t^{\boldsymbol{\Theta}})$

**Proof** [Proof of Lemma 13] Let us recall the definition of $W_1$ metric and its Kantorovich-Rubinstein duality Villani (2008), *i.e.* $W_1(\mu, \nu) \triangleq \sup_{\|g\|_{lip} \leq 1} |\mathbb{E}_{\boldsymbol{\theta} \sim \mu}[g(\boldsymbol{\theta})] - \mathbb{E}_{\boldsymbol{\theta} \sim \nu}[g(\boldsymbol{\theta})]|$. We can prove the fact that if $g(\boldsymbol{\theta}) : \mathbb{R}^d \to \mathbb{R}$ is a $L_g$-Lipschitz function in $\mathbb{R}^d$, the $g_{\boldsymbol{\Theta}}(\boldsymbol{\Theta})$, defined as $g_{\boldsymbol{\Theta}}(\boldsymbol{\Theta}) = \frac{1}{\sqrt{M}}\sum_i^M g(\boldsymbol{\theta}^{(i)})$, is a $L_g$-Lipschitz function in $\mathbb{R}^{Md}$, where $\boldsymbol{\Theta} \triangleq [\boldsymbol{\theta}^{(1)}, \cdots, \boldsymbol{\theta}^{(M)}]$:

$$\|g_{\boldsymbol{\Theta}}(\boldsymbol{\Theta}_1) - g_{\boldsymbol{\Theta}}(\boldsymbol{\Theta}_2)\| \leq \frac{1}{\sqrt{M}}\sum_{i=1}^M \|g(\boldsymbol{\theta}_1^{(i)}) - g(\boldsymbol{\theta}_2^{(i)})\| \leq \frac{L_g}{\sqrt{M}}\sum_{i=1}^M \|\boldsymbol{\theta}_1^{(i)} - \boldsymbol{\theta}_2^{(i)}\|$$

$$\leq \frac{L_g}{\sqrt{M}}\sqrt{M}\sqrt{\sum_{i=1}^M \|\boldsymbol{\theta}_1^{(i)} - \boldsymbol{\theta}_2^{(i)}\|^2} = L_g\|\boldsymbol{\Theta}_1 - \boldsymbol{\Theta}_2\|$$

Then we have:

$$\frac{1}{M}\sum_{i=1}^M \left|\mathbb{E}_{\theta_k^{(i)} \sim \mu_k}[g(\theta_k^{(i)})] - \mathbb{E}_{\boldsymbol{\theta}_t^{(i)} \sim \rho_t}[g(\boldsymbol{\theta}_t^{(i)})]\right| \overset{(1)}{=} \frac{1}{\sqrt{M}}\left|\frac{1}{\sqrt{M}}\sum_{i=1}^M (\mathbb{E}_{\theta_k^{(i)} \sim \mu_k}[g(\theta_k^{(i)})] - \mathbb{E}_{\boldsymbol{\theta}_t^{(i)} \sim \rho_t}[g(\boldsymbol{\theta}_t^{(i)})])\right|$$

$$= \frac{1}{\sqrt{M}}\left|\mathbb{E}_{\Theta_k \sim \mu_k}[g_{\boldsymbol{\Theta}}(\Theta_k)] - \mathbb{E}_{\boldsymbol{\Theta}_t \sim \rho_t}[g_{\boldsymbol{\Theta}}(\boldsymbol{\Theta}_t)]\right|$$

The (1) holds since $\mathbb{E}_{\theta_k^{(1)} \sim \mu_k}[g(\theta_k^{(1)})] = \cdots = \mathbb{E}_{\theta_k^{(M)} \sim \mu_k}[g(\theta_k^{(M)})]$ for all the particles $\theta_k^{(i)}$ and $\mathbb{E}_{\boldsymbol{\theta}_t^{(1)} \sim \rho_t}[g(\boldsymbol{\theta}_t^{(1)})] = \cdots = \mathbb{E}_{\boldsymbol{\theta}_t^{(M)} \sim \rho_t}[g(\boldsymbol{\theta}_t^{(M)})]$ for all the particles $\boldsymbol{\theta}_t^{(i)}$. Then according to the

definition of $W_1$ metric, we derive that

$$W_1(\mu_k, \rho_t) = \sup_{\|g\|_{lip} \le 1} \frac{1}{M} \sum_{i=1}^{M} \left| \mathbb{E}_{\theta_k^{(i)} \sim \mu_k}[g(\theta_k^{(i)})] - \mathbb{E}_{\boldsymbol{\theta}_t^{(i)} \sim \rho_t}[g(\boldsymbol{\theta}_t^{(i)})] \right| =$$

$$\frac{1}{\sqrt{M}} \sup_{\|g\|_{lip} \le 1} |\mathbb{E}_{\Theta_k \sim \mu_k}[g_{\boldsymbol{\Theta}}(\Theta_k)] - \mathbb{E}_{\boldsymbol{\Theta}_t \sim \rho_t}[g_{\boldsymbol{\Theta}}(\boldsymbol{\Theta}_t)]| = \frac{1}{\sqrt{M}} \sup_{\|g_{\boldsymbol{\Theta}}\|_{lip} \le 1} |\mathbb{E}_{\Theta_k \sim \mu_k}[g_{\boldsymbol{\Theta}}(\Theta_k)] - \mathbb{E}_{\boldsymbol{\Theta}_t \sim \rho_t}[g_{\boldsymbol{\Theta}}(\boldsymbol{\Theta}_t)]|$$

$$\le \frac{1}{\sqrt{M}} W_1(\mu_k^{\boldsymbol{\Theta}}, \rho_t^{\boldsymbol{\Theta}})$$

■

**Lemma 14** *Assuming $F(\mathbf{0}) = \mathbf{0}$. If $F$ in (9) is Lipschitz with constant $L_F$, and satisfies the dissipative property that $\langle F(\boldsymbol{\theta}), \boldsymbol{\theta} \rangle \ge m_F \|\boldsymbol{\theta}\|^2 - b$. Then $F_{\boldsymbol{\Theta}}$ in (28) is Lipschitz-continuous with constant $\sqrt{2}\beta^{-1}L_F + l'$ and satisfies $\langle F_{\boldsymbol{\Theta}}(\boldsymbol{\Theta}), \boldsymbol{\Theta} \rangle \ge (\beta^{-1}m_F - m')\|\boldsymbol{\Theta}\|^2 - \beta^{-1}Mb$, where $l'$ and $m'$ are some positive constants.*

**Proof** [Proof of Lemma 14]

$$\|F_{\boldsymbol{\Theta}}(\boldsymbol{\Theta}_1) - F_{\boldsymbol{\Theta}}(\boldsymbol{\Theta}_2)\| = \sqrt{\sum_i^M \|\omega_i^1 + \omega_i^2 + \omega_i^3\|^2} \le \sqrt{\sum_i^M (\|\omega_i^1\| + \|\omega_i^2\| + \|\omega_i^3\|)^2}$$

where

$$\|\omega_i^1\| = \|\beta^{-1}F(\boldsymbol{\theta}_1^{(i)}) - \beta^{-1}F(\boldsymbol{\theta}_2^{(i)})\|$$

$$\le \beta^{-1}L_F\|\boldsymbol{\theta}_1^{(i)} - \boldsymbol{\theta}_2^{(i)}\|$$

$$\|\omega_i^2\| = \|\frac{1}{M}(\sum_j^M K(\boldsymbol{\theta}_1^{(i)} - \boldsymbol{\theta}_1^{(j)})F(\boldsymbol{\theta}_1^{(j)}) - \sum_j^M K(\boldsymbol{\theta}_2^{(i)} - \boldsymbol{\theta}_2^{(j)})F(\boldsymbol{\theta}_2^{(j)}))\|$$

$$\le \|\frac{1}{M}(\sum_j^M K(\boldsymbol{\theta}_1^{(i)} - \boldsymbol{\theta}_1^{(j)})F(\boldsymbol{\theta}_1^{(j)}) - \sum_j^M K(\boldsymbol{\theta}_2^{(i)} - \boldsymbol{\theta}_2^{(j)})F(\boldsymbol{\theta}_1^{(j)}))\|$$

$$+ \|\frac{1}{M}(\sum_j^M K(\boldsymbol{\theta}_2^{(i)} - \boldsymbol{\theta}_2^{(j)})F(\boldsymbol{\theta}_1^{(j)}) - \sum_j^M K(\boldsymbol{\theta}_2^{(i)} - \boldsymbol{\theta}_2^{(j)})F(\boldsymbol{\theta}_2^{(j)}))\|$$

$$\le (2L_K H_F + L_F)\|\boldsymbol{\theta}_1^{(i)} - \boldsymbol{\theta}_2^{(i)}\|$$

$$\|\omega_i^3\| = \| - \frac{1}{M}(\sum_j^M \nabla K(\boldsymbol{\theta}_1^{(i)} - \boldsymbol{\theta}_1^{(j)}) - \sum_j^M \nabla K(\boldsymbol{\theta}_2^{(i)} - \boldsymbol{\theta}_2^{(j)}))\|$$

$$\le \frac{L_{\nabla K}}{M}(\sum_j^M \|\boldsymbol{\theta}_1^{(i)} - \boldsymbol{\theta}_2^{(i)} - (\boldsymbol{\theta}_1^{(j)} - \boldsymbol{\theta}_2^{(j)})\|)$$

$$\le L_{\nabla K}(\|\boldsymbol{\theta}_1^{(i)} - \boldsymbol{\theta}_2^{(i)}\| + \frac{1}{M}\sum_j^M \|\boldsymbol{\theta}_1^{(j)} - \boldsymbol{\theta}_2^{(j)}\|)$$

It is easy to verify that there exits some positive constant $l'$ such that

$$\|F_{\boldsymbol{\Theta}}(\boldsymbol{\Theta}_1) - F_{\boldsymbol{\Theta}}(\boldsymbol{\Theta}_2)\| = \sqrt{\sum_i^M \|\omega_i^1 + \omega_i^2 + \omega_i^3\|^2}$$

$$\le \sqrt{\sum_i^M 2(\beta^{-1}L_F + 2L_K H_F + L_F + L_{\nabla K})^2\|\boldsymbol{\theta}_1^{(i)} - \boldsymbol{\theta}_2^{(i)}\|^2 + 2\sum_j^M \|\boldsymbol{\theta}_1^{(j)} - \boldsymbol{\theta}_2^{(j)}\|^2}$$

$$\leq \sqrt{2(\beta^{-1}L_F + 2L_K H_F + L_F + L_{\nabla K})^2 + 2} \sqrt{\sum_i^M \|\boldsymbol{\theta}_1^{(i)} - \boldsymbol{\theta}_2^{(i)}\|^2}$$

$$\leq (\sqrt{2}\beta^{-1}L_F + l') \sqrt{\sum_i^M \|\boldsymbol{\theta}_1^{(i)} - \boldsymbol{\theta}_2^{(i)}\|^2}$$

$$= (\sqrt{2}\beta^{-1}L_F + l')\|\boldsymbol{\Theta}_1 - \boldsymbol{\Theta}_2\|$$

Next, we have

$$\langle F_{\boldsymbol{\Theta}}(\boldsymbol{\Theta}), \boldsymbol{\Theta} \rangle$$

$$= \sum_i^M \left( \beta^{-1} F(\boldsymbol{\theta}^{(i)})\boldsymbol{\theta}^{(i)} + \frac{1}{M}\sum_j^M K(\boldsymbol{\theta}^{(i)} - \boldsymbol{\theta}^{(j)})F(\boldsymbol{\theta}^{(j)})\boldsymbol{\theta}^{(i)} - \frac{1}{M}\sum_j^M \nabla K(\boldsymbol{\theta}^{(i)} - \theta^{(j)})\boldsymbol{\theta}^{(i)} \right)$$

Notice :

$$\sum_i^M \beta^{-1}F(\boldsymbol{\theta}^{(i)})\boldsymbol{\theta}^{(i)} \geq \beta^{-1}m_F \sum_i^M \|\boldsymbol{\theta}^{(i)}\|^2 - \beta^{-1}\beta^{-1}Mb$$

$$= \beta^{-1}m_F\|\boldsymbol{\Theta}\|^2 - \beta^{-1}Mb$$

Since we have assumed $F(0) = \mathbf{0}$, we have:

$$\sum_i^M \frac{1}{M}\sum_j^M K(\boldsymbol{\theta}^{(i)} - \boldsymbol{\theta}^{(j)})F(\boldsymbol{\theta}^{(j)})\boldsymbol{\theta}^{(i)} \geq -\frac{1}{M}\sum_i^M\sum_j^M L_F\|\boldsymbol{\theta}^{(i)}\|\|\boldsymbol{\theta}^{(j)}\|$$

$$\geq -2L_F\sum_{i=1}^M \|\boldsymbol{\theta}^{(i)}\|^2 = -2L_F\|\boldsymbol{\Theta}\|^2$$

Since $\nabla K$ is an odd function, we have:

$$\sum_i^M \frac{1}{M}\sum_j^M \nabla K(\boldsymbol{\theta}^{(i)} - \boldsymbol{\theta}^{(j)})\boldsymbol{\theta}^{(i)} \geq -\sum_i^M \frac{1}{M}\sum_j^M L_{\nabla k}\|\boldsymbol{\theta}^{(i)} - \boldsymbol{\theta}^{(j)}\|\|\boldsymbol{\theta}^{(i)}\|$$

$$\geq -3L_{\nabla K}\sum_i^M \|\boldsymbol{\theta}^{(i)}\|^2 = -3L_{\nabla K}\|\boldsymbol{\Theta}\|^2$$

As a result, we can derive the following result:

$$\langle F_{\boldsymbol{\Theta}}(\boldsymbol{\Theta}), \boldsymbol{\Theta} \rangle \geq (\beta^{-1}m - 2L_F - 3L_{\nabla K})\|\boldsymbol{\Theta}\|^2 - \beta^{-1}Mb$$

∎

Now it is ready to prove Theorem 5. It worth noting that after assuming $F(0) = \mathbf{0}$, the first bullet in Assumption 1 recovers the dissipative assumption as $\langle F(\boldsymbol{\theta}), \boldsymbol{\theta} \rangle \geq m_F\|\boldsymbol{\theta}\|^2$.

**Proof** Next we use Lemma C.5 in Xu et al. (2018) to verify that $F_{\boldsymbol{\Theta}}$ satisfies the assumptions in Raginsky et al. (2017) by setting $\delta = \frac{a'}{B}$ with $a'$ a positive constant and $B$ the size of the random set $\mathcal{I}$. Let $\mu_k^{\boldsymbol{\Theta}} := \mathcal{L}(\boldsymbol{\Theta}_k)$ and $\rho_t^{\boldsymbol{\Theta}} := \mathcal{L}(\boldsymbol{\Theta}_t)$. Now we can borrow the result of Lemma 3.6 in Raginsky et al. (2017). The relative entropy $D_{KL}(\mu_k^{\boldsymbol{\Theta}}\|\rho_{kh}^{\boldsymbol{\Theta}})$ satisfies:

$$D_{KL}(\mu_k^{\boldsymbol{\Theta}}\|\rho_{kh}^{\boldsymbol{\Theta}}) \leq (A_0\beta\frac{a'}{B} + A_1h)kh$$

with

$$A_0 = \left( 2(\frac{L_F}{\beta} + l')^2 \left( a_2 + 2(1 \vee \frac{1}{\beta^{-1}m_F - m'})(2a_1^2 + \frac{Md}{\beta}) \right) + a_1^2 \right)$$

$$A_1 = 6(\frac{L_F}{\beta} + l')^2(\beta A_0 + Md)$$

and $a_1, a_2$ are some positive constants. When the $\beta$ is small enough, there exist some positive constants $a_3, a_4$ such that

$$A_0 \leq a_3 \frac{Md}{\beta^3}, A_1 \leq a_4 \frac{Md}{\beta^4}$$

Similar to the proof of Lemma 14, it is easy to verify that there exists some positive constant $a_5$ such that $\langle F_{\boldsymbol{\Theta}}(\boldsymbol{\Theta}_1) - F_{\boldsymbol{\Theta}}(\boldsymbol{\Theta}_2), \boldsymbol{\Theta}_1 - \boldsymbol{\Theta}_2 \rangle \geq (\beta^{-1} m_F - a_5)\|\boldsymbol{\Theta}_1 - \boldsymbol{\Theta}_2\|^2$. Notice, when $\beta$ is small enough, (28) satisfies the conditions of Proposition 4.2 in Cattiaux et al. (2008). Hence, there exits some positive constant $\mathcal{C}$ such that $W_1(\mu_k^{\Theta}, \rho_{kh}^{\boldsymbol{\Theta}}) \leq \mathcal{C}\sqrt{D_{KL}(\mu_k^{\Theta}\|\rho_{kh}^{\boldsymbol{\Theta}})}$.

According to Corollary 4 and Lemma 8 in Bolley & Villani (2005), we can derive an explicit expression for $\mathcal{C}$ :

$$\mathcal{C} \leq a_6 \beta^{-1} Md \,,$$

when $\beta$ is small enough and $a_6$ is some positive constant.

Applying the Lemma 13, we have

$$W_1(\mu_k, \rho_{kh}) \leq \frac{1}{\sqrt{M}} W_1(\mu_k^{\Theta}, \rho_{kh}^{\boldsymbol{\Theta}}) \leq a_6 Md^{\frac{3}{2}}\beta^{-3}(a_3 a'\beta^2 B^{-1} + a_4 h)^{\frac{1}{2}} k^{\frac{1}{2}} h^{\frac{1}{2}}$$

Let $k = T$ and we can finish the poof. ∎

## G  PROOF OF THEOREM 6

**Proof**  Our proof is based on the proof of Lemma 3.6 in Raginsky et al. (2017) with some modifications. Firstly, adopt the same notations in the Section F and we get the following update:

$$\Theta_{k+1} = \Theta_k - \beta^{-1} G_{\mathcal{I}_k}^{\Theta} h_k + \sqrt{2\beta^{-1} h_k} \Xi_k \,, \tag{30}$$

where $\Xi_k \sim \mathcal{N}(\mathbf{0}, \mathbf{I}_{Md \times Md})$ and $h_k = \frac{h_0}{k+1}$. We assume $\mathbb{E}(G_{\mathcal{I}_k}^{\Theta}) = F_{\boldsymbol{\Theta}}(\Theta_k), \; \forall \boldsymbol{\Theta} \in \mathbb{R}^{Md}$, which is a general assumption due to the way that we choose the minibatch $\mathcal{I}_k$. We need to define $q(t)$, which will be used in the following proof:

$$q(t) = \{k \in \mathbb{R}| \sum_{i=0}^{k-1} h_i \leq t < \sum_{i=0}^{k} h_i\} \,.$$

Furthermore, we define $\sum_{i=0}^{-1} h_i \triangleq 0$ and $\sum_{i=0}^{0} h_i \triangleq h_0$ here just used for the convenience of statement in the following.

Now we focus on the following continuous-time interpolation of $\Theta_k$:

$$\boldsymbol{\Theta}(t) = \boldsymbol{\Theta}_0 - \int_0^t \tilde{G}_{\mathcal{I}(s)}^{\boldsymbol{\Theta}}\left(\boldsymbol{\Theta}(\sum_{i=0}^{q(s)-1} h_i)\right) \mathrm{d}s + \sqrt{\frac{2}{\beta}}\int_0^t \mathcal{W}_s^{(Md)},$$

where $\mathcal{I}(s) \equiv \mathcal{I}_k$ for $t \in \left[\sum_{i=0}^{k-1} h_i, \sum_{i=0}^{k} h_i\right)$, $\tilde{G}_{\mathcal{I}(s)}^{\boldsymbol{\Theta}}(\boldsymbol{\Theta}) \triangleq \frac{N}{B(s)}\sum_{q \in \mathcal{I}(s)} F_{(q)\boldsymbol{\Theta}}(\boldsymbol{\Theta})$ and $B(s)$ is the size of the minibatch $\mathcal{I}(s)$. And for each $k$, $\boldsymbol{\Theta}(\sum_{i=0}^{k-1} h_i)$ and $\Theta_k$ have the same probability law $\rho_k^{\Theta}$. Since $\boldsymbol{\Theta}(t)$ is not a Markov process, we define the following Itô process which has the same one-time marginals as $\boldsymbol{\Theta}(t)$

$$\Lambda(t) = \Theta_0 - \int_0^t \underline{G}_s\left(\Lambda(s)\right)\mathrm{d}s + \sqrt{\frac{2}{\beta}}\int_0^t \mathcal{W}_s^{(Md)}$$

$$\text{where } \underline{G}_t(x) := \mathbb{E}\left[\tilde{G}_{\mathcal{I}(t)}^{\boldsymbol{\Theta}}\left(\boldsymbol{\Theta}(\sum_{i=0}^{q(t)-1} h_i)\right)|\boldsymbol{\Theta}(t) = x\right] \,.$$

Let $\mathbf{P}_\Lambda^t := \mathcal{L}\left(\Lambda(s) : 0 \le s \le t\right)$ and $\mathbf{P}_\Theta^t := \mathcal{L}\left(\Theta(s) : 0 \le s \le t\right)$. According to the proof of lemma 3.6 in Raginsky et al. (2017), we can derive a similar result for the relative entropy of $\mathbf{P}_\Lambda^t$ and $\mathbf{P}_\Theta^t$:

$$
\begin{aligned}
D_{KL}(\mathbf{P}_\Lambda^t \,\|\, \mathbf{P}_\Theta^t) &= -\int d\,\mathbf{P}_\Lambda^t \log \frac{d\,\mathbf{P}_\Lambda^t}{d\,\mathbf{P}_\Theta^t} \\
&= \frac{\beta}{4} \int_0^t \mathbb{E}\|F_\Theta(\Lambda(s)) - \underline{G}_s(\Lambda(s))\|^2 ds \\
&= \frac{\beta}{4} \int_0^t \mathbb{E}\|F_\Theta(\Theta(s)) - \underline{G}_s(\Theta(s))\|^2 ds
\end{aligned}
$$

The last line follows that $\mathcal{L}(\Theta(s)) = \mathcal{L}(\Lambda(s))$, $\forall s$.

In the following proof, we will let $t = \sum_{i=0}^{k-1} h_i$ for some $k \in \mathbb{R}$. Now we can use the martingale property of Itô integral to derive:

$$
\begin{aligned}
&D_{KL}(\mathbf{P}_\Lambda^{\sum_{i=0}^{k-1} h_i} \,\|\, \mathbf{P}_\Theta^{\sum_{i=0}^{k-1} h_i}) \\
&= \frac{\beta}{4} \sum_{j=0}^{k-1} \int_{\sum_{i=0}^{j-1} h_i}^{\sum_{i=0}^{j} h_i} \mathbb{E}\|F_\Theta(\Theta(s)) - \underline{G}_s(\Theta(s))\|^2 ds \\
&\le \frac{\beta}{2} \sum_{j=0}^{k-1} \int_{\sum_{i=0}^{j-1} h_i}^{\sum_{i=0}^{j} h_i} \mathbb{E}\|F_\Theta(\Theta(s)) - F_\Theta(\Theta(\sum_{i=0}^{q(s)-1} h_i))\|^2 ds \\
&\quad + \frac{\beta}{2} \sum_{j=0}^{k-1} \int_{\sum_{i=0}^{j-1} h_i}^{\sum_{i=0}^{j} h_i} \mathbb{E}\|F_\Theta(\Theta(\sum_{i=0}^{q(s)-1} h_i)) - \tilde{G}_{\mathcal{I}(s)}^\Theta\left(\Theta(\sum_{i=0}^{q(s)-1} h_i)\right)\|^2 ds \\
&\le \frac{\beta L_{F_\Theta}^2}{2} \sum_{j=0}^{k-1} \int_{\sum_{i=0}^{j-1} h_i}^{\sum_{i=0}^{j} h_i} \mathbb{E}\|\Theta(s) - \Theta(\sum_{i=0}^{q(s)-1} h_i)\|^2 ds \qquad (31) \\
&\quad + \frac{\beta}{2} \sum_{j=0}^{k-1} \int_{\sum_{i=0}^{j-1} h_i}^{\sum_{i=0}^{j} h_i} \mathbb{E}\|F_\Theta(\Theta(\sum_{i=0}^{q(s)-1} h_i)) - \tilde{G}_{\mathcal{I}(s)}^\Theta\left(\Theta(\sum_{i=0}^{q(s)-1} h_i)\right)\|^2 ds \qquad (32)
\end{aligned}
$$

For the first part (31), we consider some $s \in [\sum_{i=0}^{j-1} h_i, \sum_{i=0}^{j} h_i)$. The following equation holds:

$$
\begin{aligned}
\Theta(s) - \Theta(\sum_{i=0}^{j-1} h_i) &= -(s - \sum_{i=0}^{j-1} h_i) G_{\mathcal{I}_j}^\Theta + \sqrt{2/\beta}(\mathcal{W}_s^{(Md)} - \mathcal{W}_{\sum_{i=0}^{j-1} h_i}^{(Md)}) \\
&= -(s - \sum_{i=0}^{j-1} h_i) G_{\mathcal{I}_j}^\Theta + (s - \sum_{i=0}^{j-1} h_i)(F_\Theta(\Theta_j) - G_{\mathcal{I}_j}^\Theta) \\
&\quad + \sqrt{2/\beta}(\mathcal{W}_s^{(Md)} - \mathcal{W}_{\sum_{i=0}^{j-1} h_i}^{(Md)})
\end{aligned}
$$

Thus, we can use Lemma 3.1 and 3.2 in Raginsky et al. (2017), and Lemma C.5 in Xu et al. (2018) to get the following result:

$$
\begin{aligned}
&\mathbb{E}\|\Theta(s) - \Theta(\sum_{i=0}^{j-1} h_i)\|^2 \\
&\le 3 \frac{h_0^2}{(j+1)^2} \mathbb{E}\|G_{\mathcal{I}_j}^\Theta\|^2 + 3 \frac{h_0^2}{(j+1)^2} \mathbb{E}\|F_\Theta(\Theta_j) - G_{\mathcal{I}_j}^\Theta\|^2 + \frac{6h_0 Md}{\beta(j+1)} \\
&\le 12 \frac{h_0^2}{(j+1)^2} \max_{0 \le j \le k-1}(L_{F_\Theta}^2 \mathbb{E}\|\Theta_j\|^2 + b_1) + \frac{6h_0 Md}{\beta(j+1)}
\end{aligned}
$$

where $b_1$ is some positive constant.

Consequently, the first part above (31) can be bounded as:

$$\frac{\beta L_{F_{\Theta}}^2}{2} \sum_{j=0}^{k-1} \int_{\sum_{i=0}^{j-1} h_i}^{\sum_{i=0}^{j} h_i} \mathbb{E}\|\boldsymbol{\Theta}(s) - \underline{\boldsymbol{\Theta}}(\sum_{i=0}^{q(s)-1} h_i)\|^2 \mathrm{d}s$$

$$\leq \frac{\beta L_{F_{\Theta}}^2}{2} \sum_{j=0}^{k-1} \left[ 12 \frac{h_0^3}{(j+1)^3} \max_{0 \leq j \leq K-1} (L_{F_{\Theta}}^2 \mathbb{E}\|\Theta_j\|^2 + b_1) + \frac{6h_0^2 Md}{\beta(j+1)^2} \right]$$

$$\leq \pi^2 \beta L_{F_{\Theta}}^2 h_0^3 \max_{0 \leq j \leq K-1} (L_{F_{\Theta}}^2 \mathbb{E}\|\Theta_j\|^2 + b_1) + \frac{\pi^2 L_{F_{\Theta}}^2 h_0^2 Md}{2},$$

where the last line follows from the fact that

$$\sum_{j=0}^{k-1} \frac{1}{(j+1)^3} \leq \sum_{j=0}^{k-1} \frac{1}{(j+1)^2} \leq \sum_{j=0}^{\infty} \frac{1}{(j+1)^2} = \frac{\pi^2}{6}.$$

According to Lemma C.5 in Xu et al. (2018), the second part (32) can be bounded as follows:

$$\frac{\beta}{2} \sum_{j=0}^{k-1} \int_{\sum_{i=0}^{j-1} h_i}^{\sum_{i=0}^{j} h_i} \mathbb{E}\|F_{\Theta}(\underline{\boldsymbol{\Theta}}(\sum_{i=0}^{q(s)-1} h_i)) - \tilde{G}_{\mathcal{I}(s)}^{\Theta} \left( \underline{\boldsymbol{\Theta}}(\sum_{i=0}^{q(s)-1} h_i) \right) \|^2 \mathrm{d}s$$

$$= \sum_{j=0}^{k-1} \frac{\beta h_0}{2(j+1)} \mathbb{E}\|F_{\Theta}(\Theta_j) - G_{\mathcal{I}_j}^{\Theta}\|^2$$

$$\leq \beta h_0 \max_{0 \leq j \leq k-1} (L_{F_{\Theta}}^2 \mathbb{E}\|\Theta_j\|^2 + b_1) \left( \frac{4}{B_0} + \sum_{j=1}^{k-1} \frac{4}{(j+1)(B_0 + log^{\frac{100}{99}}(j+1))} \right)$$

$$\leq \beta h_0 \max_{0 \leq j \leq k-1} (L_{F_{\Theta}}^2 \mathbb{E}\|\Theta_j\|^2 + b_1) \left( \frac{4}{B_0} + \sum_{j=1}^{k-1} \frac{4}{(j+1)log^{\frac{100}{99}}(j+1)} \right)$$

$$\leq (b_2 + \frac{4}{B_0})\beta h_0 \max_{0 \leq j \leq k-1} (L_{F_{\Theta}}^2 \mathbb{E}\|\Theta_j\|^2 + b_1),$$

where the last line follows from the fact that when $r > 1$,

$$\sum_{j=1}^{k-1} \frac{4}{(j+1)\log^r(j+1)} \leq \sum_{j=1}^{\infty} \frac{4}{(j+1)\log^r(j+1)} \leq \frac{4\log^{1-r} 2}{r-1}.$$

Denote $\mu_k^{\Theta} := \mathcal{L}(\Theta_k)$ and $\rho_t^{\Theta} := \mathcal{L}(\boldsymbol{\Theta}_t)$. Due to the data-processing inequality for the relative entropy, we have

$$D_{KL}(\mu_k^{\Theta} \| \rho_{\sum_{i=0}^{k-1} h_i}^{\Theta}) \leq D_{KL}(\mathbf{P}_{\Lambda}^{\sum_{i=0}^{k-1} h_i} \| \mathbf{P}_{\Theta}^{\sum_{i=0}^{k-1} h_i})$$

$$\leq \pi^2 \beta L_{F_{\Theta}}^2 h_0^3 \max_{0 \leq j \leq k-1} (L_{F_{\Theta}}^2 \mathbb{E}\|\Theta_j\|^2 + b_1) + \frac{\pi^2 L_{F_{\Theta}}^2 h_0^2 Md}{2}$$

$$+ (b_2 + \frac{4}{B_0})\beta h_0 \max_{0 \leq j \leq k-1} (L_{F_{\Theta}}^2 \mathbb{E}\|\tilde{\boldsymbol{\Theta}}_j\|^2 + b_1)$$

$$\leq (\pi^2 \beta L_{F_{\Theta}}^2 h_0^3 + b_2 \beta h_0 + \frac{4}{B_0}\beta h_0) \max_{0 \leq j \leq k-1} (L_{F_{\Theta}}^2 \mathbb{E}\|\Theta_j\|^2 + b_1) + \frac{\pi^2 L_{F_{\Theta}}^2 h_0^2 Md}{2}.$$

Lemma 3.2 in Raginsky et al. (2017) has provided a uniform bound to $\max_{0 \leq j \leq k-1} (L_{F_{\Theta}}^2 \mathbb{E}\|\Theta_j\|^2 + b_1)$. Hence we can tell that $D_{KL}(\mathbf{P}_{\Lambda}^{\sum_{i=0}^{k-1} h_i} \| \mathbf{P}_{\Theta}^{\sum_{i=0}^{k-1} h_i})$ would not increase w.r.t. $k$. This is a nice property that the fixed-step-size SPOS does not endow. Since $L_{F_{\Theta}} = \sqrt{2}\beta^{-1} L_F + l'$, it is easy to

verify that when $\beta$ is small enough, there exists some positive constants $b_3, b_4, b_5$ and $b_6$ such that:

$$
D_{KL}(\mu_k^{\Theta} \| \rho_{\sum_{i=0}^{k-1} h_i}^{\Theta})
$$
$$
\leq (\pi^2 \beta L_{F_\Theta}^2 h_0{}^3 + b_2 \beta h_0 + \frac{4\beta h_0}{B_0}) \max_{0 \leq j \leq K-1} (L_{F_\Theta}^2 \mathbb{E}\|\tilde{\Theta}_j\|^2 + b_1) + \frac{\pi^2 L_{F_\Theta}^2 h_0{}^2 Md}{2}
$$
$$
\leq (b_3 h_0^3 + \frac{b_4 \beta^3 h_0}{B_0} + b_5 h_0^2 \beta^2) \frac{Md}{\beta^4} \; .
$$

Similar to the proof of Theorem 5, we can bound the $W_1(\mu_k^{\Theta} \| \rho_{\sum_{i=0}^{k-1} h_i}^{\Theta})$ term with Corollary 4, Lemma 8 in Bolley & Villani (2005) and Proposition 4.2 in Cattiaux et al. (2008). Specifically, when $\beta$ is small enough, there exist some positive constant $a_6$ such that:

$$
W_1(\mu_k^{\Theta} \| \rho_{\sum_{i=0}^{k-1} h_i}^{\Theta}) \leq a_6 (\frac{Md}{\beta}) \sqrt{D_{KL}(\mu_k^{\Theta} \| \rho_{\sum_{i=0}^{k-1} h_i}^{\Theta})}
$$
$$
\leq a_6 \beta^{-3} M^{\frac{3}{2}} d^{\frac{3}{2}} (b_3 h_0^3 + \frac{b_4 \beta^3 h_0}{B_0} + b_5 h_0^2 \beta^2)^{\frac{1}{2}} \; .
$$

According to Lemma 13, we have

$$
W_1(\mu_k, \rho_{kh}) \leq \frac{1}{\sqrt{M}} W_1(\mu_k^{\Theta} \| \rho_{\sum_{i=0}^{k-1} h_i}^{\Theta}) = a_6 \beta^{-3} M d^{\frac{3}{2}} (b_3 h_0^3 + \frac{b_4 \beta^3 h_0}{B_0} + b_5 h_0^2 \beta^2)^{\frac{1}{2}}
$$

∎

Let $k = T$ and we can finish the proof.

# H   DETAILED STATEMENTS OF NONASYMPTOTIC CONVERGENCE UNDER THE CONVEX CASE

We give detailed statements of non-asymptotic convergence of SPOS under the convex case, which can be proved by directly combining results from Theorem 3–6, thus they are omitted for simplicity. We consider a fixed-stepsize and a decreasing-stepsize cases in Theorem 15 and Theorem 16, respectively.

**Theorem 15 (Fixed Stepsize)** *Under Assumption 1, if we set $h_k = h_0$ and $B_k = B_0$, we can bound the $W_1(\mu_T, \nu_\infty)$ as*

$$
W_1(\mu_T, \nu_\infty) \leq \frac{c_1}{\sqrt{M}(\beta^{-1} - c_2)} + c_3 \exp\left\{ -2\left(\beta^{-1} m_F - 3H_F L_K - 2L_F\right) Th \right\}
$$
$$
+ c_3 M d^{\frac{3}{2}} \beta^{-3} (c_4 \beta^2 B^{-1} + c_5 h)^{\frac{1}{2}} T^{\frac{1}{2}} h^{\frac{1}{2}} \; .
$$

*where $(c_1, c_2, c_3, c_4, c_5, c_6, \beta)$ are some positive constants such that $\frac{1}{\beta} > c_2$ and $\frac{m_F}{\beta} > 3H_F L_K - 2L_F$.*

**Theorem 16 (Decreasing Stepsize)** *Under Assumption 1, if we set $h_k = h_0/(k+1)$ and $B_k = B_0 + [\log(k+1)]^{100/99}$, we can bound the $W_1(\mu_T, \nu_\infty)$ as*

$$
W_1(\mu_T, \nu_\infty) \leq \frac{c_1}{\sqrt{M}(\beta^{-1} - c_2)} + c_3 \exp\left\{ -2\left(\beta^{-1} m_F - 3H_F L_K - 2L_F\right) (\sum_{k=0}^{T-1} h_k) \right\}
$$
$$
+ c_3 \beta^{-3} M d^{\frac{3}{2}} (c_6 h_0^3 + \frac{c_7 \beta^3 h_0}{B_0} + c_8 h_0^2 \beta^2)^{\frac{1}{2}} \; .
$$

*where $(c_1, c_2, c_3, c_6, c_7, c_8, \beta)$ are some positive constants such that $\frac{1}{\beta} > c_2$ and $\frac{m_F}{\beta} > 3H_F L_K - 2L_F$, as*

# I   PROOF OF THEOREM 7 AND COROLLARY 8

**Proof** [Proof of Theorem 7]

$$
\mathrm{d}\left(\boldsymbol{\theta}_t^{(i)} - \boldsymbol{\theta}_t^{(j)}\right) = -\beta^{-1}\left(F(\boldsymbol{\theta}_t^{(i)}) - F(\boldsymbol{\theta}_t^{(j)})\right)\mathrm{d}t
$$

$$
+ \frac{1}{M}\sum_q^M\left[\nabla K(\boldsymbol{\theta}_t^{(i)} - \boldsymbol{\theta}_t^{(q)}) - \nabla K(\boldsymbol{\theta}_t^{(j)} - \boldsymbol{\theta}_t^{(q)})\right]\mathrm{d}t
$$

$$
- \frac{1}{M}\sum_q^M\left(F(\boldsymbol{\theta}_t^{(q)})K(\boldsymbol{\theta}_t^{(i)} - \boldsymbol{\theta}_t^{(q)}) - F(\boldsymbol{\theta}_t^{(q)})K(\boldsymbol{\theta}_t^{(j)} - \boldsymbol{\theta}_t^{(q)}))\right)\mathrm{d}t
$$

$$
+ \sqrt{\frac{2}{\beta}}(\mathrm{d}\mathcal{W}_t^{(i)} - \mathrm{d}\mathcal{W}_t^{(j)})
$$

$$
\Rightarrow \mathrm{d}\left(\mathbb{E}\sum_{ij}^M\left\|\boldsymbol{\theta}_t^{(i)} - \boldsymbol{\theta}_t^{(j)}\right\|^2\right) = -\mathbb{E}2\sum_{ij}^M\beta^{-1}\left(F(\boldsymbol{\theta}_t^{(i)}) - F(\boldsymbol{\theta}_t^{(j)})\right)\left(\boldsymbol{\theta}_t^{(i)} - \boldsymbol{\theta}_t^{(j)}\right)\mathrm{d}t
$$

$$
+ \mathbb{E}\sum_{ij}^M\frac{2}{M}\sum_q^M\left[\nabla K(\boldsymbol{\theta}_t^{(i)} - \boldsymbol{\theta}_t^{(q)}) - \nabla K(\boldsymbol{\theta}_t^{(j)} - \boldsymbol{\theta}_t^{(q)})\right]\left(\boldsymbol{\theta}_t^{(i)} - \boldsymbol{\theta}_t^{(j)}\right)\mathrm{d}t
$$

$$
- \mathbb{E}\sum_{ij}^M\frac{2}{M}\sum_q^M\left(F(\boldsymbol{\theta}_t^{(q)})K(\boldsymbol{\theta}_t^{(i)} - \boldsymbol{\theta}_t^{(q)}) - F(\boldsymbol{\theta}_t^{(q)})K(\boldsymbol{\theta}_t^{(j)} - \boldsymbol{\theta}_t^{(q)}))\right)\left(\boldsymbol{\theta}_t^{(i)} - \boldsymbol{\theta}_t^{(j)}\right)\mathrm{d}t
$$

$$
+ \mathbb{E}2\sum_{ij}^M\sqrt{\frac{2}{\beta}}(\mathrm{d}\mathcal{W}_t^{(i)} - \mathrm{d}\mathcal{W}_t^{(j)})\left(\boldsymbol{\theta}_t^{(i)} - \boldsymbol{\theta}_t^{(j)}\right)
$$

$$
\leq -2\beta^{-1}m_F\mathbb{E}\sum_{ij}^M\left\|\boldsymbol{\theta}_t^{(i)} - \boldsymbol{\theta}_t^{(j)}\right\|^2\mathrm{d}t - 2m_K\mathbb{E}\sum_{ij}^M\left\|\boldsymbol{\theta}_t^{(i)} - \boldsymbol{\theta}_t^{(j)}\right\|^2\mathrm{d}t
$$

$$
+ 2H_F L_K\mathbb{E}\sum_{ij}^M\left\|\boldsymbol{\theta}_t^{(i)} - \boldsymbol{\theta}_t^{(j)}\right\|^2\mathrm{d}t
$$

$$
+ 2\sqrt{\frac{2}{\beta}}\left(\mathbb{E}\sum_{ij}^M(\mathrm{d}\mathcal{W}_t^{(i)} - \mathrm{d}\mathcal{W}_t^{(j)})^2\right)^{1/2}\left(\mathbb{E}\sum_{ij}^M\left\|\boldsymbol{\theta}_t^{(i)} - \boldsymbol{\theta}_t^{(j)}\right\|^2\right)^{1/2}.
$$

Denote $z(t) = \mathbb{E}\sum_{ij}^M\left\|\boldsymbol{\theta}_t^{(i)} - \boldsymbol{\theta}_t^{(j)}\right\|^2$. We have

$$
z(t)' \leq -(2\beta^{-1}m_F + 2m_K - 2H_F L_K)z(t) + 4M\sqrt{\frac{d}{\beta}z(t)} \tag{33}
$$

We can finish our proof by applying Gronwall Lemma on (33).  ∎

**Proof** [Proof for Corollary 8]

$$
\mathrm{d}\left(\boldsymbol{\theta}_t^{(i)} - \boldsymbol{\theta}_t^{(j)}\right) =
$$

$$
+ \frac{1}{M}\sum_q^M\left[\nabla K(\boldsymbol{\theta}_t^{(i)} - \boldsymbol{\theta}_t^{(q)}) - \nabla K(\boldsymbol{\theta}_t^{(j)} - \boldsymbol{\theta}_t^{(q)})\right]\mathrm{d}t
$$

$$
- \frac{1}{M}\sum_q^M\left(F(\boldsymbol{\theta}_t^{(q)})K(\boldsymbol{\theta}_t^{(i)} - \boldsymbol{\theta}_t^{(q)}) - F(\boldsymbol{\theta}_t^{(q)})K(\boldsymbol{\theta}_t^{(j)} - \boldsymbol{\theta}_t^{(q)}))\right)\mathrm{d}t
$$

$$\Rightarrow \mathrm{d}\left(\mathbb{E}\sum_{ij}^{M}\left\|\boldsymbol{\theta}_t^{(i)}-\boldsymbol{\theta}_t^{(j)}\right\|^2\right)=$$

$$+\mathbb{E}\sum_{ij}^{M}\frac{2}{M}\sum_{q}^{M}\left[\nabla K(\boldsymbol{\theta}_t^{(i)}-\boldsymbol{\theta}_t^{(q)})-\nabla K(\boldsymbol{\theta}_t^{(j)}-\boldsymbol{\theta}_t^{(q)})\right]\left(\boldsymbol{\theta}_t^{(i)}-\boldsymbol{\theta}_t^{(j)}\right)\mathrm{d}t$$

$$-\mathbb{E}\sum_{ij}^{M}\frac{2}{M}\sum_{q}^{M}\left(F(\boldsymbol{\theta}_t^{(q)})K(\boldsymbol{\theta}_t^{(i)}-\boldsymbol{\theta}_t^{(q)})-F(\boldsymbol{\theta}_t^{(q)})K(\boldsymbol{\theta}_t^{(j)}-\boldsymbol{\theta}_t^{(q)}))\right)\left(\boldsymbol{\theta}_t^{(i)}-\boldsymbol{\theta}_t^{(j)}\right)\mathrm{d}t$$

$$\leq -2m_K\mathbb{E}\sum_{ij}^{M}\left\|\boldsymbol{\theta}_t^{(i)}-\boldsymbol{\theta}_t^{(j)}\right\|^2\mathrm{d}t+2H_FL_K\mathbb{E}\sum_{ij}^{M}\left\|\boldsymbol{\theta}_t^{(i)}-\boldsymbol{\theta}_t^{(j)}\right\|^2\mathrm{d}t$$

Denote z(t)= $\mathbb{E}\sum_{ij}^{M}\left\|\boldsymbol{\theta}_t^{(i)}-\boldsymbol{\theta}_t^{(j)}\right\|^2$. We have

$$z(t)' \leq -(2m_K-2H_FL_K)z(t) \tag{34}$$

We can finish our proof by applying Gronwall Lemma on (34). ∎

## J    PROOF OF THEOREM 9

**Proof** [Proof of Theorem 9] Our conclusion for $\tilde{\mathcal{B}}(\hat{\mu}_T,\hat{\mu}_\infty)$ is essentially a specification of the result in Mattingly et al. (2002), which has also been applied in Xu et al. (2018).

Specifically, we rely on the following lemma, which is essentially Theorem 7.3 in Mattingly et al. (2002) and Lemma C.3 in Xu et al. (2018), considering the SDE (16):

$$\mathrm{d}\boldsymbol{\Theta}_t=-F_{\boldsymbol{\Theta}}(\boldsymbol{\Theta}_t)\mathrm{d}t+\sqrt{2\beta^{-1}}\mathrm{d}\mathcal{W}_t^{(Md)}$$

As mentioned in Section 5, we firstly denote the distribution of $\boldsymbol{\Theta}_t$ as $\rho_t^{\boldsymbol{\Theta}}$. Then we define the $\hat{\Theta}_k \triangleq [\hat{\theta}_k^{(1)},\cdots,\hat{\theta}_k^{(M)}] \in \mathbb{R}^{Md}$, which is actually the numerical solution of (16) using full gradient with Euler method. And we denote the distribution of $\hat{\Theta}_k$ as $\hat{\mu}_k^{\boldsymbol{\Theta}}$.

**Lemma 17** *Let $F_{\boldsymbol{\Theta}}$ be Lipschitz-continuous with constant $L_{\boldsymbol{\Theta}}$, and satisfy the dissipative property that $\langle F_{\boldsymbol{\Theta}}(\boldsymbol{\Theta}),\boldsymbol{\Theta}\rangle \geq m_{\boldsymbol{\Theta}}\|\boldsymbol{\Theta}\|^2-b_{\boldsymbol{\Theta}}$. Define $V_{\boldsymbol{\Theta}}(\boldsymbol{\Theta})=C_0+L_{\boldsymbol{\Theta}}/2\|\boldsymbol{\Theta}\|^2$. The Euler method for (16) has a unique invariant measure $\hat{\mu}_\infty^{\boldsymbol{\Theta}}$, and for all test function $f_{\boldsymbol{\Theta}}$ such that $|f_{\boldsymbol{\Theta}}|\leq V_{\boldsymbol{\Theta}}(\boldsymbol{\Theta})$, we have*

$$\left|\mathbb{E}[f_{\boldsymbol{\Theta}}(\hat{\Theta}_k))]-\mathbb{E}_{\hat{\Theta}_\infty\sim\hat{\mu}_\infty^{\boldsymbol{\Theta}}}[f(\hat{\Theta}_\infty)]\right| \leq C\kappa\rho^{-Md/2}(1+\kappa e^{m_{\boldsymbol{\Theta}}h})\exp\left(-\frac{2m_{\boldsymbol{\Theta}}kh\rho^{Md}}{\log(\kappa)}\right),$$

*where $\rho\in(0,1)$, $C>0$ are positive constants, and $\kappa=2L_{\boldsymbol{\Theta}}(b_{\boldsymbol{\Theta}}\beta+m_{\boldsymbol{\Theta}}\beta+Md)/m_{\boldsymbol{\Theta}}$.*

Now we adopt the $f_{\boldsymbol{\Theta}}:\mathbb{R}^{Md}\to\mathbb{R}$ defined as $f_{\boldsymbol{\Theta}}(\boldsymbol{\Theta})=\frac{1}{M}\sum_i^M f(\boldsymbol{\theta}^{(i)})$, where $f:\mathbb{R}^d\to\mathbb{R}$ is a $L_f$-Lipschitz function and satisfies our Assumption 2 and $\boldsymbol{\Theta}\triangleq[\boldsymbol{\theta}^{(1)},\cdots,\boldsymbol{\theta}^{(M)}]$. Similar to the proof of Lemma 12, we can find that $f_{\boldsymbol{\Theta}}:\mathbb{R}^{Md}$ is a $L_f/\sqrt{M}$-Lipschitz function. Furthermore, according to the Lemma 14, it is easily check that $F_{\boldsymbol{\Theta}}$ is $L_{\boldsymbol{\Theta}}$-Lipschitz where $L_{\boldsymbol{\Theta}}=\sqrt{2}\beta^{-1}L_F+l'$. Hence, when the $\beta$ is small enough, we have $L_f/\sqrt{M}\leq\sqrt{2}\beta^{-1}L_F+l'$. Then we can set the $C_0$ large enough to force $f_{\boldsymbol{\Theta}}$ to satisfy the condition in Lemma 17 that $|f_{\boldsymbol{\Theta}}|\leq V_{\boldsymbol{\Theta}}(\boldsymbol{\Theta})$. Then, according to the exchangeability of the particle system $\{\hat{\theta}_k^{(i)}\}$ and Lemma 14, we can bound the $\tilde{\mathcal{B}}(\hat{\mu}_T,\hat{\mu}_\infty)$ as

$$\tilde{\mathcal{B}}(\hat{\mu}_T,\hat{\mu}_\infty)\leq\left|\mathbb{E}[f_{\boldsymbol{\Theta}}(\hat{\Theta}_T))]-\mathbb{E}_{\hat{\Theta}_\infty\sim\hat{\mu}_\infty^{\boldsymbol{\Theta}}}[f(\hat{\Theta}_\infty)]\right|$$

$$\leq C_2\varsigma\sigma^{-Md/2}(1+\varsigma e^{m_{\boldsymbol{\Theta}}h})\exp\left(-2m_{\boldsymbol{\Theta}}Th\sigma^{Md}/\log(\varsigma)\right)$$

where $\varsigma = 2L_{\Theta}(Mb\beta + m_{\Theta}\beta + Md)/m_{\Theta}$, $L_{\Theta} = \sqrt{2}\beta^{-1}L_F + l'$, $m_{\Theta} = \beta^{-1}m - m'$, and $(\sigma, C_2, l', m')$ are some positive constants independent of (T, M, h) and $\sigma \in (0, 1)$.

To prove the bound for $\tilde{\mathcal{B}}(\hat{\mu}_{\infty}, \rho_{\infty})$ Since $\hat{\Theta}_k = (\hat{\theta}_k^{(1)}, \cdots, \hat{\theta}_k^{(M)})$ can be considered as a solution to the SDE (16), standard results from linear FP equation can be applied. Specifically, for the $\tilde{\mathcal{B}}(\hat{\mu}_{\infty}, \rho_{\infty})$ term, we rely on the following lemma adapted from Lemma C.4 in Xu et al. (2018)Chen et al. (2015), and is essentially the work of Chen et al. (2015) when taking $K \to \infty$.

**Lemma 18** *Under the same assumption as in Lemma 17, for the Lipschitz-continuous function* $f_{\Theta}(\Theta) = \frac{1}{M}\sum_i^M f(\theta^{(i)})$ *mentioned above, the following bound is satisfied for some positive constant C:*

$$\left| \frac{1}{K}\sum_{k=1}^{K-1} \mathbb{E}[f_{\Theta}(\hat{\Theta}_k)] - \mathbb{E}_{\Theta_{\infty} \sim \rho_{\infty}^{\Theta}}[f(\Theta_{\infty})] \right| \leq C(\frac{h}{\beta} + \frac{\beta}{Kh}) \,.$$

The uniqueness of invariant measure of the Euler method from Lemma 17 implies the numerical solution $\hat{\Theta}_k$ to be ergodic. Then similar to the proof of Lemma 4.2 in Xu et al. (2018), we consider the case where $K \to \infty$. Take average over the $\{\hat{\Theta}_k\}_{k=0}^{K=1}$, we have

$$\mathbb{E}_{\hat{\Theta}_{\infty} \sim \hat{\mu}_{\infty}^{\Theta}}[f_{\Theta}(\hat{\Theta}_{\infty})] = \lim_{K \to \infty} \frac{1}{K}\sum_{k=1}^{K-1} \mathbb{E}[f_{\Theta}(\hat{\Theta}_k)]$$

Now according to the exchangeability of the particle system $\{\hat{\theta}_k^{(i)}\}$ and $\{\theta_t^{(i)}\}$, we can bound the $\tilde{\mathcal{B}}(\hat{\mu}_{\infty}, \rho_{\infty})$ as :

$$\tilde{\mathcal{B}}(\hat{\mu}_{\infty}, \rho_{\infty}) \leq \left| \mathbb{E}_{\hat{\Theta}_{\infty} \sim \hat{\mu}_{\infty}^{\Theta}}[f_{\Theta}(\hat{\Theta}_{\infty})] - \mathbb{E}_{\Theta_{\infty} \sim \rho_{\infty}^{\Theta}}[f_{\Theta}(\Theta_{\infty})] \right| \leq C_3 h/\beta$$

where $C_3$ are some positive constant. ∎

## K  PROOF OF THEOREM 10

**Proof** [Proof of Theorem 10] Adopting the same notation used in the proof of the Theorem 5, we define $\Theta_k \triangleq [\theta_k^{(1)}, \cdots, \theta_k^{(M)}]$ and $G_{\mathcal{I}_k}^{\Theta} \triangleq \frac{N}{B_k}\sum_{q \in \mathcal{I}_k} F_{(q)\Theta}(\Theta_k)$. We denote the distribution of $\Theta_k$ as $\mu_k^{\Theta}$.

$$\Theta_{k+1} = \Theta_k - \beta^{-1}G_{\mathcal{I}_k}^{\Theta}h_k + \sqrt{2\beta^{-1}h_k}\Xi_k \,,$$

We firstly give a bound to the $W_2(\mu_k^{\Theta}, \hat{\mu}_k^{\Theta})$ (the definition of the $\hat{\mu}_k^{\Theta}$ has been mention in the last section). According to the proof of the Lemma 4.4 in Xu et al. (2018)

$$W_2(\mu_k^{\Theta}, \hat{\mu}_k^{\Theta}) \leq kh(L_{\Theta}\Gamma' + MC_4)\left((6 + 2\Gamma')\beta/B\right)^{1/2}$$

where $\Gamma' = 2(1 + 1/m_{\Theta})(Mb + 2M^2C_4^2 + Md/\beta)$ and $C_4$ is some positive constant independent of (T, M, h). Note the fact that $W_1(\mu_k^{\Theta}, \hat{\mu}_k^{\Theta}) \leq W_2(\mu_k^{\Theta}, \hat{\mu}_k^{\Theta})$ and $W_1(\mu_k, \hat{\mu}_k) \leq \frac{1}{\sqrt{M}}W_1(\mu_k^{\Theta}, \hat{\mu}_k^{\Theta})$ (see the proof of Lemma 13, similar result holds here), We get

$$W_1(\mu_T, \hat{\mu}_T) \leq Th(L_{\Theta}\Gamma' + MC_4)\left((6 + 2\Gamma')\beta/(BM)\right)^{1/2} \,.$$

Let us compare the definition of $\mathcal{W}_1(\mu, \nu)$ and $\tilde{\mathcal{B}}(\mu, \nu)$

$$W_1(\mu, \nu) \triangleq \sup_{\|g\|_{lip} \leq 1} |\mathbb{E}_{\theta \sim \mu}[g(\theta)] - \mathbb{E}_{\theta \sim \nu}[g(\theta)]|$$

$$\tilde{\mathcal{B}}(\mu, \nu) \triangleq |\mathbb{E}_{\theta \sim \mu}[f(\theta)] - \mathbb{E}_{\theta \sim \nu}[f(\theta)]|$$

and we can derive the result that $\tilde{\mathcal{B}}(\mu_T, \hat{\mu}_K) \leq L_f W_1(\mu_T, \hat{\mu}_T)$. Now we finish our proof. ∎

## L    DISCUSSION ON THE COMPLEXITY OF OUR METHOD

The complexity of an algorithm mainly refers to its time complexity (corresponding to the number of iterations in our method *i.e. T*) and space complexity (corresponding to the number of particles used in our method *i.e. M*). Hence the complexity of our method can be well explored with our work, since our non-asymptotic convergence theory was developed w.r.t. the number of particles *i.e. M* and iterations *i.e. T*. Their relationship (tradeoff) was even discussed  further in Experiment 6.1. Moreover, comparing (9) with (3) , one can easily find that our space complexity is exactly the same as SVGD and our computational time in each iteration is almost the same as SVGD with an extra addition operation. However, it worth noting that our method have much better performance in practice and no "pitfall" verified by both our theory and our experiments. .

## M    COMPARISON WITH RELATED WORK

Firstly, our proposed framework SPOS is different from the recently proposed particle-optimization sampling framework (Chen et al., 2018), in the sense that we solve the nonlinear PDE (6) stochastically. For example they deterministically solve the equation in (6) $\partial \nu_t = \beta^{-1} \nabla_{\boldsymbol{\theta}} \cdot \nabla_{\boldsymbol{\theta}} \nu_t$ approximately using blob method adopted from (Carrillo et al., 2017).

Secondly, our method is also distinguishable to existing work on granular media equations such as (Durmus et al., 2018). Their work about the Granular media equations focuses on the following PDE:

$$\partial_t \nu_t = \nabla_{\boldsymbol{\theta}} \cdot \left( \nu_t \beta^{-1} F(\boldsymbol{\theta}) + \nu_t \left( \nabla K * \nu_t(\boldsymbol{\theta}) \right) + \beta^{-1} \nabla_{\boldsymbol{\theta}} \nu_t \right) \ , \tag{35}$$

whereas our framework focuses on the following one:

$$\partial_t \nu_t = \nabla_{\boldsymbol{\theta}} \cdot \left( \nu_t \beta^{-1} F(\boldsymbol{\theta}) + \nu_t \left( E_{Y \sim \nu_t} K(\boldsymbol{\theta} - Y) F(Y) - \nabla K * \nu_t(\boldsymbol{\theta}) \right) + \beta^{-1} \nabla_{\boldsymbol{\theta}} \nu_t \right) \ . \tag{36}$$

The extra term $\nu_t \left( E_{Y \sim \nu_t} K(\boldsymbol{\theta} - Y) F(Y) \right)$ in our framework makes the analysis much more challenging. The main differences are summarized below:

- Formulations are different. The extra term $E_{Y \sim \mu_t} K(\boldsymbol{\theta} - Y) F(Y)$ cannot be combined with the $F(\boldsymbol{\theta})$ term in their (35). This is because function $F(\boldsymbol{\theta})$ **itself** is a function independent of $t$; while $E_{Y \sim \mu_t} K(\boldsymbol{\theta} - Y) F(Y)$ depends on both $\boldsymbol{\theta}$ and $t$. This makes our problem much more difficult.

- Assumptions are different.  For example, the analysis on granular media equations in (Cattiaux et al., 2008) requires that $F$ satisfies a special condition $C(\mathbb{A}, \alpha)$, which is a strong condition impractical to be satisfied in our case; And Durmus et al. (2018) adopts different assumptions from ours with a different goal.

- For the Euler integrator, Durmus et al. (2018) does not consider an Euler solution. Furthermore, our sampling method needs "stochastic gradient" *i.e.* $G_k^{(i)} \triangleq \frac{N}{B_k} \sum_{q \in \mathcal{I}_k} F_q(\theta_k^{(i)})$ in (9) for computational feasibility, which is quite different from the former work on particle-SDE such as (Malrieu, 2003; Cattiaux et al., 2008). Few of the former work on particle-SDE considered the stochastic gradient issue.

To sum up, the main purpose of our paper is to provide a non-asymptotic analysis of our method instead of improving the former work on a certain type of PDE. This is also the reason why we said that part of our proof techniques are based on those for analyzing granular media equations.

## N    EXTRA EXPERIMENTS

### N.1    TOY EXPERIMENTS

We compare the proposed SPOS with other popular methods such as SVGD and standard SGLD on four mutil-mode toy examples.  We aim to sample from four unnormalized 2D densities $p(z)/exp\{U(z)\}$, with the functional form provided in Rezende & Mohamed (2015). We optimize/sample 50 and 2000 particles to approximate the target distributions. The results are illustrated in Figure 4 and Figure 5, respectively.

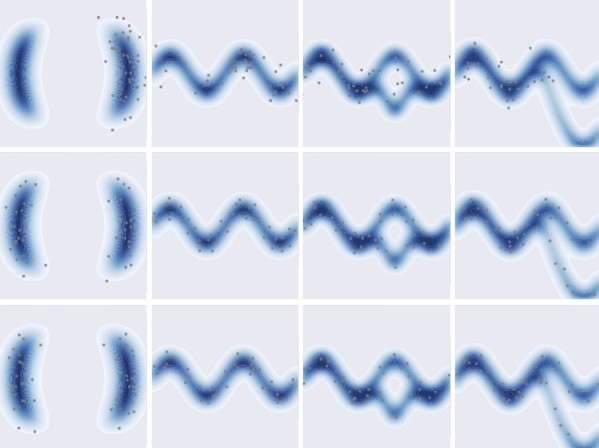

Figure 4: Illustration of different algorithms on toy distributions. Dots are the final particles; the blue regions represent ground true densities. Each column is a distribution case. First row: standard SGLD; Second row: SVGD; Third row: SPOS.

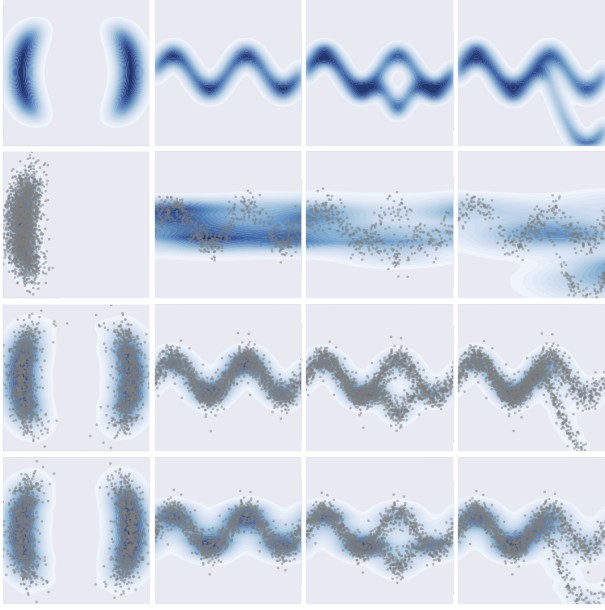

Figure 5: Illustration of different algorithms on toy distributions. Dots are the final particles; the blue regions represent densities estimated by the particles. Each column is a distribution case. First row: ground true densities; Second row: standard SGLD; Third row: SVGD; Fourth row: SPOS.

### N.1.1  BAYESIAN NEURAL NETWORKS FOR MNIST CLASSIFICATION

We perform the classification tasks on the standard MNIST dataset. A two-layer MLP 784-X-X-10 with ReLU activation function is used, with X being the number of hidden units for each layer. The training epoch is set to 100. The test errors are reported in Table 2. Surprisingly, the proposed SPOS outperforms other algorithms such as SVGD at a significant level, though it is just a simple modification of SVGD by adding in random Gaussian noise. This is partly due to the fact that our SPOS algorithm can jump out of local modes efficiently, as explained in Section 4.4.

Table 2: Classification error of FNN on MNIST.

| Method | Test Error | |
|---|---|---|
| | 400-400 | 800-800 |
| SPOS | **1.32%** | **1.24%** |
| SVGD | 1.56% | 1.47% |
| SGLD | 1.64% | 1.41% |
| RMSprop | 1.59% | 1.43% |
| RMSspectral | 1.65% | 1.56% |
| SGD | 1.72% | 1.47% |
| BPB, Gaussian | 1.82% | 1.99% |
| SGD, dropout | 1.51% | 1.33% |

