# OpenReview forum: "Towards More Theoretically-Grounded Particle Optimization Sampling for Deep Learning"
_ICLR.cc/2019/Conference_

### Official Review · AnonReviewer3 · 2018-11-02
**I have multiple concerns regarding proof of Theorem 3**

**Rating:** 3
**Confidence:** 4

**Review:**

This paper considers the problem of Bayesian inference using particle optimization sampler. Similarly to SGLD, authors propose Stochastic Particle Optimization Sampler (SPOS), augmenting Stein Variational Gradient Descent (SVGD) with diminishing Gaussian noise, replacing the hard-to-compute term of the Chen et al. (2018) formulation. Various theoretical results are given.

This paper was a pleasant read until I decided to check the proof of Theorem 3. I was not able to understand transitions in some of the steps and certain statements in the proof seem wrong.

Theorem 3:
"Note that $\theta^i_t$ and $\hat \theta^i_t$ are initialized with the same initial distribution µ0 = ν0 and we can also set $\theta^i_0$ to be independent of $\hat \theta^i_0$, we can have $\gamma(0) = 0$. $\gamma(0) = E \|\theta^i_0 - \hat \theta^i_0 \|^2$." - this doesn't seem right to me. Expectation of squared difference of two independent and identically distributed random variables is not 0, assuming expectation is with respect to their joint density.

"Then according to the Gronwall Lemma, we have" - I don't see how the resulting inequality was obtained. When I tried applying Gronwall Lemma, it seems that authors forgot to multiply by $t$ and  $\lambda_1$. Could you please elaborate how exactly Gronwall Lemma was used in this case.

"... some positive constants c1 and c2 independent of (M, d)$ - in the proof authors introduce additional assumption "We can tune the bandwidth of the RBF kernel to make ∇K ≤ H_∇K, which is omitted in the Assumption due to the space limit." First, there is a missing norm, since ∇K is a vector and H_∇K is I believe a scalar constant. Second, c1 = H_∇K + H_F, which both bound norm of d-dimensional vector and hence depend on d. I also suggest that all assumptions are included in the theorem statements, especially since authors have another assumption requiring large bandwidth. Additionally, feasibility of these both assumptions being satisfied should be explored (it seems to me that they can hold together, but it doesn't mean that part of assumptions can be moved to the supplement).

I find using Wasserstein-1 metric misleading in the theorem statement . This is not what authors really bound - from the proof it can be seen that they bound W_1 with W_2 and then with just an expectation of l2 norm. Moreover I don't understand the meaning of this bound. Theorem is concerned with W_1 distance between two atomic measures. What is the expectation over? Note that atom locations are supposed to be fixed for the W_1 to make sense in this context (and the expectation is over the coupling of discrete measures defined by weights of the atoms, not atom locations).

"Note the first bullet indicates U to be a convex function and W to be ... " I think it should be K, not W.

Theorems 3-6 could be lemmas, while there should be a unifying theorem for the bound.

Finally, I think notation should be changed - same letter is used for Wasserstein distance and Wiener process.

Other comments:

Example in Figure 1 is somewhat contrived - clearly gradient based particle sampler will never escape the mode since all modes are disconnected by regions with 0 density. Proposed method on the other hand will eventually jump out due to noise, but it doesn't necessarily mean it produces better posterior estimate. Something more realistic like a mixture of Gaussians, with density bounded away from zero across domain space, will be more informative.

It is not sufficient to report RMSE and test log likelihood for BNNs. One of the key motivating points is posterior uncertainty estimation. Hence important metric, when comparing to other posterior inference techniques, is to show high uncertainty for out of distribution samples and low for training/test data.

---

> ### Author Response · Authors · 2018-11-09
> **Other minor issues**
>
> 1. We will change the “W” to “K”.
>
> 2. Yes, we understand your suggestions that Theorems 3-6 could be lemmas and there should be a unifying theorem for the bound. However, it is worth noting that the “unifying theorems for the bounds” are provided in the appendix H as mentioned at the end of  Section 4.3. And we hope not to move them to the context due to the space limit. Our paper is already 10 pages long and can not fit these two long theorems. Besides, we have no problems to change the “Theorem” to “Lemma”. Unfortunately, as our responses to other reviewers, our paper’s main contribution lies in the theoretical analysis. We think the techniques and ideas of Theorems 3-6 are also very important, which provide some guides for other researchers in the field. Hence, we think Theorems 3-6 worth the name “Theorem”, not just “Lemmas” (which mean they are only affiliated to the “unifying theorems for the bounds”). But if you still disagree with our opinions, we are willing to make the changes since this issue is quite minor and should not be the reason for your rejection.
>
> 3. We will change the notation of Wasserstein metric from $\mathcal{W}$ to $W$.
>
> 4. We don’t think “Example in Figure 1 is contrived”. The distribution form used in Figure 1 is given in Appendix A. It is obvious that the distribution are nonzero everywhere (the probabilities are just very small somewhere), thus it does not have the problem of “disconnected modes”. We use this example to show failure case of SVGD, which induces no problem with our SPOS.
>
> Using RMSE and log-likelihood is the gold standard in Bayesian learning of DNNs. Instead of directly showing uncertainty for in/out distribution samples as suggested by you, we test It in the more direct scenario of reinforcement learning. The reason is that it is well-accepted that RL performance directly measures how well the uncertainty is learned, as there is an exploration stage in the learning, requiring uncertainty to explore the environment. As a result, we believe our measure in the experiments are standard.
>
> At last, we really hope you could reconsider your scoring as it seems to be quite unfair based on your comments. In our response, we have fully address the minor problem which you pointed out. Besides, we have resolved your concerns about the W_1 metric excessively. And we have shared our opinions with you on the name of our Theorem and decides to change the notations as you suggested.
>
> We think it is really unfair to reject our paper merely for some minor problems, which has been fully addressed. Although we have explained much to your concerns about W_1 metric, most of them are not needed to add to our paper, which means that we do no need to make much revision. If you like, we are willing to add our explanations for your concerns into the Appendix. Thank you so much for your time and re-consideration!

---

> ### Author Response · Authors · 2018-11-09
> **Misunderstandings of the Wasserstein metric (Part 3)**
>
> 3. Until now, you might have another concern: “why do you choose W_1 instead of W_2?” Although many work on SG-MCMC adopted W_2 metric, it is worth noting that providing a bound for SPOS is much more complicated than for SGLD. Hence, we think focusing on W_1 metric is quite acceptable. Moreover, W_1 is adopted also because of the work on the asymptotic convergence analysis of SVGD in Liu, 2017.
>
> The asymptotic convergence analysis of SVGD in Liu, 2017 adopts “BL metric”. The definition of BL metric is $BL(\mu,\nu) \triangleq sup{E f(\mu) - E f(\nu), || f ||_{\infty} <=1 and || f ||_{Lip}<=1}$. And please notice that W_1 metric has another well-know definition, $W_1(\mu,\nu) \triangleq sup{E f(\mu) - E f(\nu), || f ||_{Lip}<=1}$. Hence, the definitions of these metrics are similar but not identical. (Actually it is easy to verify that $BL(\mu,\nu) <= W_1(\mu,\nu) $. ) Therefore, we decide to adopt W_1 metric here due to its similarity to BL metric in the existing work.

---

> ### Author Response · Authors · 2018-11-09
> **Misunderstandings of the Wasserstein metric (Part 2.2)**
>
> 2)The W_1 or W_2 metrics between $\mu_k$ and the posterior distribution are also broadly used in the convergence analysis of SG-MCMC and non-convex optimization. This is the second reason.
>
> Similarly, the ${\theta}_{k}$ in the updates of SG-MCMC is actually also a random variable. And we denote its distribution as $\mu_k$. It is worth noting that in practice, SG-MCMC also collects many “fixed atom locations” one by one, and they also form a discrete distribution. But many recent work on SG-MCMC also only care about $\mu_k$ and use W_1 or W_2 metric to make convergence analysis, such as Xu et al. (2018), Raginsky et al. (2017) in our reference and other related papers like “On the Theory of Variance Reduction for Stochastic Gradient Monte Carlo”(ICML 2018), “Further and stronger analogy between sampling and optimization: Langevin Monte Carlo and gradient descent”(COLT 2017) and “User-friendly guarantees for the Langevin Monte Carlo with inaccurate gradient”(arXiv:1710.00095). Hence, we decided to follow what the existing work does.
>
> Another point we want to emphasize is that Bayesian sampling algorithms like SGLD are more and more broadly used in non-convex optimization. In non-convex optimization, the “complicated stochastic discrete distribution” is much less useful than $\mu_k$. Please refer to Xu et al. (2018), Raginsky et al. (2017) in our reference for more details. And they also adopt W_1 or W_2 metric between $\mu_k$ and their target distribution in their analysis. It is worth noting that we have pointed out in Section 7, the Conclusion, that non-convex optimization is one of the interesting future work for our SPOS. Hence, it is much more valuable to give a bound in terms of the W_1 metric between $\mu_k$ and our target distribution.

---

> ### Author Response · Authors · 2018-11-09
> **Misunderstandings of the Wasserstein metric (Part 2.1)**
>
> 2.We think we could know your other potential concern from your comments.
>
> Although we feel sorry that from your comments, it is too hard for us to get exactly whtat you mean, but we still try our best to guess your concern from your wording, like “discrete measures” and “atom locations”. We are afraid that your concern might not merely exists in your understanding of Theorem 3, but also makes you confused throughout the whole paper. Hence, we decide to use our paper’s main target, bounding the W_1 metric between $\mu_k$ and the posterior distribution, mentioned at the beginning of Section 4.1 to resolve your concern (we use $\mu_k$ instead of $\mu_T$ here just for the sake of clarity).
>
> Before we present your concerns, please look at equation (9). Due to the fact that $\xi_{k-1}^(i)$ are random variables, the particle ${\theta}_{k}^{(i)}$ in our SPOS are also random variables. In other words, ${\theta}_{k}^{(i)}$, for any i and k, is a random variable and has its own distribution. Just as what we mentioned in Section 4.1, due to the exchangeability of those particles, if we initialize them with the same distribution $\mu_0$, all the particles ${\theta}_{k}^{(i)}$ will have the same distribution for any k, denoted as $\mu_k$.
>
> Now, we will present your concern as follows. We guess you might focus on the issue that in practice, what we get is several “fixed atom locations” of the particles after running our SPOS algorithm. And based on those “fixed atom locations”, we will get a “discrete distribution”, not the $\mu_k$ as we mentioned above. What’s worse, due to the fact that each particle ( ${\theta}_{k}^{(i)}$ ) is a random variable, the “fixed atom locations” do not remain the same. This leads to the problem that the “discrete distribution” we derive becomes stochastic, making our problem much more complicated and the “expectation” extremely hard for you to understand.
>
> However, please notice our paper do not focus on that “complicated stochastic discrete distribution”. What we bound in our paper is the W_1 metric between $\mu_k$ and our target distribution (posterior distribution), where $\mu_k$ is the distribution of each ${\theta}_{k}^{(i)}$ instead of that “complicated stochastic discrete distribution”. This has been mentioned in our beginning of Section 4.1.
>
> Until now, we guess you might have the question why we do not work on that “complicated stochastic discrete distribution”. We have the following two reasons:
>
> 1)The W_1 metric between $\mu_k$ and the posterior distribution is adopted due to the goal of Bayesian sampling. This is our first reason.
>
> In our Bayesian sampling algorithm SPOS, the particles ${\theta}_{k}^{(i)}$ (or we can call them atoms) are actually “parameters”, which are used to characterize the corresponding statistical models in Bayesian statistics. (Please see the first sentence of our Section 2.1 which provides the basic background of Bayesian sampling). Those “parameters” ${\theta}_{k}^{(i)}$ all have the same distribution of $\mu_k$. And since the target of Bayesian sampling algorithm is to sample some parameters from the posterior distribution, how $\mu_k$ approximates the posterior distribution is exactly what we need to work on. Therefore, we do not need to care about that “complicated changing discrete distribution”.

---

> ### Author Response · Authors · 2018-11-09
> **Misunderstandings of the Wasserstein metric (Part 1)**
>
> As for your concerns on Wasserstein-1 metric, we are afraid that there might be a lot of misunderstandings.
>
> 1.Your first concern is about the correctness of our way of bounding W_1.
>
> First of all, please notice what we want to bound in Theorem 3 is the W_1 distance between $\rho_t$ and $\nu_t$.
>
> Next, please notice that, for every t, the particles $\theta_t^{(i)}$ in Equation (8) and the proof of Theorem 3 are *random variables*. In our theoretical analysis, they are not “fixed atom locations” as you mentioned due to the Wiener process in Equation (8). And as mentioned at the beginning of Section 4.1, “due to the exchangeability of the particle system $\{\theta_t^{(i)}\}_{i=1}^M$ in Equation (8), if we initialize all the particles $\theta_t^{(i)}$ with the same distribution $\rho_0$, they would endow the same distribution for each time $t$. We denote the distribution of each $\theta_t^{(i)}$ as $\rho_t$.” Hence, the $\rho_t$ in our statement of Theorem 3 is actually the distribution of each $\theta_t^{(i)}$.
>
> Similar results hold for $\nu_t$. According to the definition of $\bar{\theta}_t^{(i)}$, the $\nu_t$ in our statement of Theorem 3 is the distribution of each $\bar{\theta}_t^{(i)}$.
>
> Now let’s explain why we “bound W_1 with W_2 and then with just an expectation of l2 norm” as you mentioned. In Equation (26) of the proof for Theorem 3, the expectation “E || \theta_{t}^{(i)} - \bar{\theta}_t^{(i)} ||^2” is taken over the distribution of the coupling $( \theta_{t}^{(i)} , \bar{\theta}_t^{(i)} )$. And the distribution of $( \theta_{t}^{(i)} , \bar{\theta}_t^{(i)} )$ is a joint distribution with marginal distributions equal to $\rho_t$ and $\nu_t$. According to the definition of W_2 distance mentioned at the beginning of Section 4, we can bound W_2 “with just an expectation of l2 norm”.
>
> To sum up, there is no problem bounding W_1 by bounding W_2, which is achieved by bounding the expectation term. And we are so sorry to say that we do not need to make changes to the proof since all the definitions used in the proof have been provided in our context. And the order of these definitions has been set carefully for reading.

---

> ### Author Response · Authors · 2018-11-09
> **Typo and minor flaw which have been fully addressed and do not affect the correctness of our theorem**
>
> 1. There is a typo in the presentation of Gronwell Lemma. The last line of Lemma 12 should be changed to “v(t) \leq  v(a) \exp (\int_{a}^{t}\beta(s)\mathrm{d}s)”. We missed the $\exp$ here. We recommend you to read this page https://en.wikipedia.org/wiki/Gr%C3%B6nwall%27s_inequality if you need more information about this basic mathematics knowledge.
>
> 2. Yes, you are correct. When we set $\theta_0^{(i)}$ to be independent of $\bar{\theta}_0^{(i)}$, the $\gamma_i(t) \triangleq \mathbb{E}\left\|\theta_t^{(i)} - \bar{\theta}_t^{(i)}\right\|^2$ does not equal to zero.
>
> However, we can address your concern. Actually, we can set $\bar{\theta}_0^{(i)}$ identical to $\theta_0^{(i)}$, which will then make the $\gamma_i(0)$ equal to zero! “$\bar{\theta}_0^{(i)}$ identical to $\theta_0^{(i)}$” means that when $\theta_{0}^{(i)}$ equals to some value like $\theta_0^{(i)}=x$, the $\bar{\theta}_0^{(i)}$ will also equals to $x$, where $x$ is some real number. In other words, the statement “we can set $\bar{\theta_{0}}^{(i)}$ to be independent of $\theta_{0}^{(i)}$ " should be changed to ”we can set $\bar{\theta}_{0}^{(i)}$ identical to $\theta_0^{(i)}$”.
>
> We guess you might immediately have the following question, “why can you set $\bar{\theta}_{0}^{(i)}$ identical to $\theta_0^{(i)}$”. Yes, we can. Please notice the fact that $\bar{\theta}_t^{(i)}$ is introduced only for our proof convenience, it is user-defined, just as former work on analyzing granular media equations in Malrieu (2003); Cattiaux et al. (2008); Durm us et al. (2018). That said, the $\bar{\theta}_t^{(i)}$ does not exist in our interpretation of our algorithm. Theoretically, we can let $\bar{\theta}_0^{(i)}$ satisfy any distribution. But most of them are not useful for our proof. To prove Theorem 3 (we can change its name to “Lemma” as your suggestions), we choose to set it identical to $\theta_0^{(i)}$. The introduction of $\bar{\theta}_t^{(i)}$ is broadly used in former literature of both Mathematics and Machine Learning like Malrieu (2003); Cattiaux et al. (2008); Durm us et al. (2018). if you still have concerns, we strongly recommend you to scan those literature, which we have cited.
>
> 3. Until now, we have addressed the minor flaw. Based on the above explanations, we will show you the correctness of Theorem 3 in detail, which means we will show you how Gronwell Lemma works here.
>
> First, please locate the following statement in our proof “\Rightarrow	(\sqrt{\gamma(t)}-\frac{(H_{\nabla K}+H_F)/\sqrt{2}}{\sqrt{M}(\beta^{-1}-3H_FL_K-2L_F)})^\prime    \leq      -\lambda_1  (\sqrt{\gamma(t)}-\frac{(H_{\nabla K}+H_F)/ \sqrt{2}}{\sqrt{M}(\beta^{-1}-3H_FL_K-2L_F)})”.    (Finding the rightarrow in our proof will help you!)
>
> Now applying the Gronwell Lemma, we can derive that:
> \begin{align*}
> \sqrt{\gamma(t)} -   \frac {(H_{\nabla K}+H_F)/ \sqrt{2}}    {\sqrt{M}(\beta^{-1}-3H_FL_K-2L_F)}    \leq
> \(  \sqrt{\gamma(0)}  -   \frac  {(H_{\nabla K}+H_F)/ \sqrt{2}}  {\sqrt{M}(\beta^{-1}-3H_FL_K-2L_F)} \)    \exp(-\lambda_1 t)
> \end{align*}
>
> Next, it is worth noting that $ \exp(-\lambda_1 t) < 1$ since $\lambda_1> 0$ and $t> 0$. And according to $\gamma(0)}=0$, we get that
> \begin{align*}
> \sqrt{\gamma(t)}   \leq     \frac  {(H_{\nabla K}+H_F)/\sqrt{2}}   {\sqrt{M}(\beta^{-1}-3H_FL_K-2L_F)}
> \end{align*}
> ,which is what we need.
>
> Until now we have addressed your concerns about the proof of our theorem 3.
>
> 4. We appreciate your suggestions on the additional assumptions, we will include them in the assumption statements. And we agree with your statement that c1 depends on d, and we will rephrase it.

---

> ### Author Response · Authors · 2018-11-09
> **Thank you for the detailed review.**
>
> Thanks for your detailed review! We can see that you went to some details of our paper and applied a higher standard to it, which we are really grateful. Unfortunately, there appears to be lots of misunderstandings. We hope you would not mind that we decide to explain in detail to address your concerns. Despite the long and detailed explanation, we only need to revise our paper slightly to fully address your comments.
>
> We divide the whole rebuttal into several parts, which we believe will be easier for you to follow. We hope you can read through our rebuttal even if you encounter some new questions during reading. We believe your new questions can also be addressed after finishing reading our rebuttal. We hope you could read them carefully, and we think our rebuttal is also very helpful for other researchers who read this paper. Thank you.

---

> > ### Comment · AnonReviewer3 · 2018-11-10
> > **This rebuttal, although contains useful pieces, is unprofessional and largely irrelevant to the review**
> >
> > Authors made mistakes in their proof and even the theorem statement itself needs corrections. Despite admitting to it in the subsequent comments, in the summary statement they say "unfortunately, there appears to be lots of misunderstandings". The only "misunderstanding" as far as I can tell is with regard to the distribution under the expectation bounding Wasserstein distance. I am also surprised to hear that authors think I applied "higher standard" to their paper. Checking proofs of the theorems in a theoretically oriented paper appears as a basic standard to me.
> >
> > All that was needed from the rebuttal was to admit the mistakes and suggest resolutions when possible. Although authors partially did so, they decided to spend bulk of the rebuttal inventing some new questions I did not ask and answering to themselves. I suggest you delete the irrelevant parts of your rebuttal.
> >
> > Regarding your reading recommendation - If the authors expect the reader to consult Wikipedia about the Gronwell Lemma, perhaps they should include the link in the manuscript instead of presenting this "basic mathematics knowledge" with a typo. And if such typo was made, it is best to simply admit and correct it. In my review I said that "authors forgot to multiply by $t$ and  $\lambda_1$". Surely if we include $\exp$ in the Lemma formulation, this missing terms are found as $\exp(-\lambda_1 t) < 1$ in your rebuttal.
> >
> > Regarding "misunderstanding" - I agree that your bound on W2 is correct and follows from the definition of W2 when expectation is with respect to the optimal coupling. Although finding such coupling could be interesting for discrete measures with random atoms (unlike well-known cases of discrete optimal transport), but looks like it is not needed for your proof. Nonetheless I recommend to add a brief explanation to the distribution under expectation under the bound and why (if so) you don't need to construct it.
> >
> > Regarding W1 and W2: indeed I decided to check your proof because I wanted to see how you handle W1 (as opposed to W2, which is known to be easier to analyze). Your proof does not bring anything new for someone interested in studying W1 since you are essentially working with W2. Hence I think that using W1 metric in your theorem statements is misleading and unnecessary.
> >
> > To summarize, I recommend that the mistakes are corrected and manuscript is resubmitted to a journal or a future ML venue as it needs to undergo additional round of review. My recommendation remains to be "reject" at this point.

---

> > > ### Author Response · Authors · 2018-11-14
> > > **Respectfully, we hope to make the following points**
> > >
> > > Thank you for your response. We apologize for the previous long rebuttal. Nonetheless, we didn’t mean to write an “unprofessional” rebuttal, but hope to provide all the details and to solve possible doubts one might encounter when reading the rebuttal. We respect your decision, but still want to make the following points.
> > >
> > > 1. First of all, as pointed out by our rebuttal, we can revise one line in the presentation of Gronwell Lemma and the minor flaw in our proof of Theorem 3 (by changing “independent” to “identical”) to correct the proof. The only minor “correction” we can find in the “statement” is changing “(M,d)” to “(M,t)”, which does not affect the correctness of our result in Theorem 3. By the way, we have admitted the unintended typo and have corrected it in our rebuttal.
> > >
> > > 2. We are glad that you have resolved your previous confusions such as the “distribution under the expectation bounding Wasserstein distance”, and you also agree that our bound on W2 is correct. That is just what our rebuttal, “misunderstandings of the Wasserstein metric (Part 2.1 & 2.2)”, aims at. Respectfully, we do not think our necessary rebuttal is “largely irrelevant to the review” since it helps eliminate possible questions in your original review. For example, in your original review you mentioned “discrete measures defined by weights of the atoms, not atom locations”; you also said you don't understand the meaning of this bound and Theorem is concerned with W_1 distance between two “atomic measures”. Hence, we didn’t mean to invent new questions (actually we didn’t), but tried to provide detailed explanations.
> > >
> > >
> > > 3. Actually when we wrote our rebuttal, we knew your potential misunderstandinig which might be summarized by the word “unnecessary” in your latest response. That is the reason why we decided to provide our rebuttal, “ misunderstandings of the Wasserstein metric (Part 3)”. We have emphasized there that choosing W_1 instead of W_2 in the statement Theorem 3 is also due to similarity to the work on the asymptotic convergence analysis of SVGD in Liu, 2017.
> > >
> > > Besides, we want to emphasize the main propose of our paper aims at developing the first non-asymptotic convergence theory for SVGD-style algorithm, SPOS. We are not inventing new approaches to deal with W_1 distance. Respectfully, we think it is unreasonable to say that our use of W_1 distance in the statement is “misleading” the researchers who are looking for new approaches to deal with W_1. We did not try to invent the new approaches indeed, and have never mentioned that inventing the new approaches is our aim in our paper.
> > >
> > > 4. We always respect your decision. We were just a little disappointed that the reason for rejection is due to misunderstanding or typo and minor flaw (which have been addressed in our rebuttal).
> > >
> > > Again, thank you for your latest response which provides us another chance to re-emphasize the importance of our rebuttal. Please let us know if you still have questions. We are ready to answer them respectfully.

---

### Official Review · AnonReviewer2 · 2018-11-02

**Rating:** 4
**Confidence:** 3

**Review:**

This paper proposes a particle-based inference algorithm, the optimal update for each particle is the summation of the standard SGLD direction and SVGD velocity.  The work further analyzes non-asymptotic properties of SPOS. The results appear theoretically interesting and of potential practical value in designing inference algorithms. I did not go through the proofs in the supplementary.

[Experimental results are not convincing]

[BNN] I noticed the test RMSE and test LL of SVGD are directly copied from the original SVGD paper. However, the performance critically depends on:
1.    Running time, or training epochs
2.    Data partitions
To be a fair comparison, the authors should keep at least the training epochs and random partitions the same. Especially for the dataset Year, for which only one random partition is conducted. It’s highly likely that the performance gain is due to favored data partition rather than the superiority of the algorithm.

[RL] Average rewards are significantly lower than the scores reported in the original SVPG paper?
1.    From figure 3, SPOS only outperforms SVPG on envs Cartpole Swing Up and Double Pendulum. The best reward for env Cartpole Swing Up reported in this paper is around 200. However, the score is ~400 in the original SVPG paper. For the env Double Pendulum, there’s also very large performance gap. I am aware the code for SVPG is now publicly available, the authors may consider conducting the experiments with the same settings (e.g. same seed?). Otherwise, it’s hard to tell whether the performance gain is significant while the baseline is much worse than it should be.
2.    Only 3 envs are reported, the authors may also consider reporting all the envs are used in the SVPG paper

[Figure 1] The authors may consider reporting the exact settings of this case, otherwise, it’s hard to believe that SVGD would collapse on a simple 1D case.

If the authors can fully address the concerns above, I will consider changing the scores.

Other comments:

-    Related papers:
     Stein Variational Message Passing for Continuous Graphical Models,  Wang et al., ICML18 (https://arxiv.org/abs/1711.07168)
     Stein Variational Gradient Descent as Moment Matching, Liu et al., NIPS18 (https://arxiv.org/abs/1810.11693)

-    Page 30 crashes my browser all the time

---

> ### Author Response · Authors · 2018-11-07
> **Thanks for your comments**
>
> To eliminate the confusion of the reviewer, we re-run the experiments for SVGD and SPOS, the same split of data (train, val and test) are used for SVGD and SPOS. The test results are reported on the best model on the validation set. The results are as follows:
>
> Boston_housing:
> SPOS (MSE)         & SVGD (MSE)        &  SPOS (LL)              & SVGD (LL)
> 2.829 \pm 0.126 & 2.961 \pm 0.109 &  -2.532 \pm 0.082 & -2.591 \pm 0.029
>
> Concrete:
> SPOS (MSE)           & SVGD (MSE)        & SPOS (LL)              & SVGD (LL)
> 5.071 \pm 0.1495 & 6.157 \pm 0.082 & -3.062 \pm 0.037 & -3.247 \pm 0.01
>
> Energy:
> SPOS (MSE)           & SVGD (MSE)        & SPOS (LL)              & SVGD (LL)
> 0.752 \pm 0.0285 & 1.291 \pm 0.029 & -1.158 \pm 0.073 & -1.534 \pm 0.026
>
> Kin8nm:
> SPOS (MSE)         & SVGD (MSE)        & SPOS (LL)            & SVGD (LL)
> 0.079 \pm 0.001 & 0.075 \pm 0.001 & 1.092 \pm 0.013 & 1.138 \pm 0.004
>
> Naval:
> SPOS (MSE)     & SVGD (MSE)        & SPOS (LL)          & SVGD (LL)
> 0.004 \pm 0.0 & 0.004 \pm 0.000 & 4.145 \pm 0.02 & 4.032 \pm 0.008
>
> CCPP:
> SPOS (MSE)           & SVGD (MSE)        & SPOS (LL)              & SVGD (LL)
> 3.939 \pm 0.0495 & 4.127 \pm 0.027 & -2.794 \pm 0.025 & -2.843 \pm 0.006
>
> Winequality:
> SPOS (MSE)         & SVGD (MSE)        & SPOS (LL)              & SVGD (LL)
> 0.598 \pm 0.014 & 0.604 \pm 0.007 & -0.911 \pm 0.041 & -0.926 \pm 0.009
>
> Yacht:
> SPOS (MSE)         & SVGD (MSE)        & SPOS (LL)              & SVGD (LL)
> 0.84 \pm 0.0865 & 1.597 \pm 0.099 & -1.446 \pm 0.121 & -1.818 \pm 0.06
>
> Protein:
> SPOS (MSE)         & SVGD (MSE)        &  SPOS (LL)             & SVGD (LL)
> 4.254 \pm 0.005 & 4.392 \pm 0.015 & -2.876 \pm 0.009 & -2.905 \pm 0.010
>
>
> For the YearPredict data, we follow the literature, and only report one result (the training is quite stable for this dataset).
>
> For RL results, the four benchmarks are the simplest benchmarks for reinforcement learning; thus it is obviously not necessary to use a 400-400 MLP as a policy. Even in much more complex benchmarks, e.g., humanoid and walker, previous methods such as soft-Q learning and SAC used 128-128 MLP or 256-256 MLP as the policy network, TRPO used one-layer MLP. We followed the settings of VIME, and think it is more reasonable to use a simpler policy network. 400-400 MLP as a policy is too complex to be a good choice. We used the released code of SVPG and the same settings for both methods (also same seeds), thus the comparisons are fair for both methods.
>
> For the other environments, Mountain car is a very simple environment compared with CartpoleSwingUp and Double Pendulum, and we encountered errors from the framework when running the algorithms. We will try to fix the problem and incorporate results into our next revision.
>
> As in our response to Reviewer 1, we did not claim a better algorithm than SVGD in theory because there is no nonasymptotic theory for SVGD (though we did observed better empirical performance), but a better way to understand the nonasymptotic convergence behavior of particle optimization algorithms, e.g., SPOS, providing a non-asymptotic bound for an SVGD-style algorithm the first time.

---

### Official Review · AnonReviewer1 · 2018-11-04
**a hybrid SGLD - SVGD**

**Rating:** 5
**Confidence:** 4

**Review:**

Two promising methods for scalable sample-based Bayesian inference are:
1) SGLD: simply discretize a standard Langevin dynamics to construct a Markov chain that approximate the correct invariant distribution. This reads:

x_{t+1} = x_t + \nabla \log \pi(x_t) \delta + \sqrt(2 \delta) \xi

2) SVGD: the method can be expressed as a type of gradient descent of an appropriate functional on the space of probability distributions. A cloud of particles {x_i}_{I=1}^M evolves according to:

x^i_{t+1} = x^i_t + (some functional of all the particles) \, \delta

The method proposed in the article is not very different from alternating the two above mentioned update, which is indeed quite a natural idea, and can work pretty well I think. The method reads:

x^i_{t+1} = x^i_t + [ \nabla \log \pi(x_t) \delta + (some functional of all the particles) \, \delta ] + \sqrt(2 \delta) \xi.

PROS:
- yes, I think that the method can work quite OK since it may be borrowing the strengths of both SGLD and SVGD.
- It seems that the meat of the paper consists in proving some (non-asymptotic) convergence result. Unfortunately, this went above my head and I cannot claim that I have read the details of the proofs.

CONS:
- it is (very) difficult to fairly evaluate this type of methods in high-dimensional settings. I thus appreciate that the numerical section starts with a toy very simple Gaussian model. I would have been much more interested  in fair and extensive simulations in this type of settings where it is relatively easy to compare the proposed method with SGLD and SGVD. In other words, after reading the paper, I must say that I am not at all convinced that the method does bring something over SGLD or SVGD (although it is very possible that it does).  For example, comprehensive and fair comparisons with SGLD and SVGD  in Gaussian settings (not necessarily one-dimensional) could have been presented. The delicate tuning of the different methods, the speed of convergence wrt algorithmic time, the speed of comparison wrt the number of particles, etc.. could have been investigated numerically: this would have been, I think, much more convincing.

MINOR comments:
- I did check the proof of Theorem 2, which seems hand-wavy and overly complicated.  What is the function G? It seems that the proof of Theorem 2 simply consists in establishing that if each particle x_i follows the dynamics dx = F(x)*dt then the associated densities satisfy \partial_t \mu_t = -\partial_x(F(x) * \mu_t(x))  , which is obvious. But the situation in the paper is indeed more delicate since the particles are interacting, etc... Reading this proof got me very worried and did not motivate me to read the rest of the paper.

SUMMARY:
- the method is not terribly original -- this is a simple hybrid SVGD / SGLD -- but may work very well.
- unfortunately, the numerical experiments are not convincing.

---

> ### Author Response · Authors · 2018-11-06
> **we don't think the comments are fair**
>
> We thank the reviewer for his comments, however, the reviewer seems to miss our key point.
>
> First, we would like to stress that our paper does not try to show our proposed method is better than either SVGD or SGLD. The motivation of our method is to help better understand the nonasymptotic convergence behavior of SVGD. Since there is no/limited nonasymptotic theory for SVGD, it is hard to understand its convergence behavior. To overcome this difficulty, we combine SGLD with SVGD, and for the first time successfully develop nonasymptotic convergence theory for a SVGD-style algorithm. Because there no nonasymptotic theory for SVGD (except for some restrict results of the recent work [1]), nothing can be said about SVGD and our algorithm in theory. Similarly, it is hard to compare to SGLD as well because our algorithm is particle based.
>
> That said, even though we can perform other experiments on simple toy data, nothing can be expected by comparing our algorithm and SVGD, except the pitfall property of SVGD described in Sec 4.4, which has been shown in Figure 1.
>
> For the proof of Theorem 2, G is defined in the equation below eq.20, which is related to LHS of eq.7. It is unfair to say that "Reading this proof got me very worried and did not motivate me to read the rest of the paper" because the proof techniques of Theorem 2 and other theorems are complete independent.

---

### Meta-Review · Area_Chair1 · 2018-12-14
**Interesting paper but Improvement and Clarification are needed**

**Confidence:** 5
**Recommendation:** Reject

**Metareview:**

This paper proposes a combination of SVGD and SLGD and analyzes its non-asymptotic properties based on gradient flow. This is an interesting direction to explore. Unfortunately, two major concerns have been raised regarding this paper:  1) the reviewers identified multiple technical flaws. Authors provided rebuttal and addressed some of the problems. But the reviewers think it requires significantly more improvement and clarification to fully address the issues. 2) the motivation of the combination of SVGD and SLGD, despite of being very interesting, is not very clearly motivated; by combining SVGD and SLGD, one get convergence rate for free from the SLGD part, but not much insight is shed on the SVGD part (meaning if the contribution of SLGD is zero, then the bound because vacuum). This could be misleading given that one of the claimed contribution is non-asymptotic theory of ''SVGD-style algorithms" (rather than SLGD style..). We encourage the authors to addresses the technical questions and clarify the contribution and motivation of the paper in revision for future submissions.